# The Cylindrical Representation Hypothesis for Language Model Steering

Lang Gao [* 1]  Jinghui Zhang [* 1]  Wei Liu [2]  Fengxian Ji [1]  Chenxi Wang [1]
Zirui Song [1]  Akash Ghosh [1]  Youssef Mohamed [1]  Preslav Nakov [1]  Xiuying Chen [1]

## Abstract

Steering is widely used for controlling large language models, yet its effects are often unstable and difficult to predict. Existing theoretical accounts are largely based on the Linear Representation Hypothesis (LRH), which assumes that concepts can be orthogonalized for lossless control. However, this assumption rarely holds in practice and cannot explain the variability of steering outcomes. We propose the *Cylindrical Representation Hypothesis* (CRH), a geometric extension of LRH that relaxes the orthogonality assumption while preserving linear concept representations. We show that overlapping concept contributions naturally induce a sample-specific cylindrical structure consisting of a ***central axis***, a ***normal plane***, and ***sensitive sectors***. The central axis captures the primary semantic transition associated with a target concept, while the normal plane governs steering sensitivity. Within this plane, some sectors facilitate concept activation, while others suppress or delay it. CRH reveals an asymmetry in steering predictability: the normal plane can be inferred from difference vectors, but the sensitive sectors cannot, introducing an intrinsic source of uncertainty. This explains why steering outcomes vary across samples even when intervention directions are well aligned. Experiments spanning 100 concepts, multiple models, and diverse steering methods provide consistent evidence for the predicted cylindrical structure, suggesting that steering variability arises from representation geometry rather than imperfect steering vectors. Our code is available at https://github.com/mbzuai-nlp/CRH.

*Equal contribution [1]Department of Natural Language Processing, Mohamed bin Zayed University of Artificial Intelligence, Abu Dhabi, UAE [2]Department of Computer Science, National University of Singapore, Singapore. Correspondence to: Xiuying Chen <Xiuying.Chen@mbzuai.ac.ae>.

*Proceedings of the 43rd International Conference on Machine Learning*, Seoul, South Korea. PMLR 306, 2026. Copyright 2026 by the author(s).

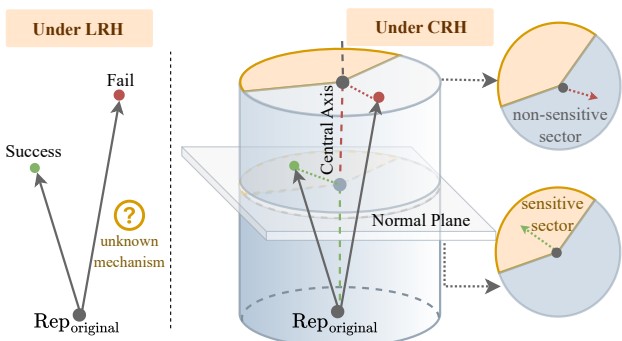

*Figure 1.* **Comparison of LRH and CRH.** LRH assumes a single global concept direction, while CRH reveals a sample-specific cylindrical structure with a central axis and an orthogonal normal plane. Steering outcomes depend on the sector within the normal plane, exposing the sample-specific nature of steerability.

## 1. Introduction

As large language models (LLMs) become more capable, researchers have become increasingly interested in understanding their internal mechanisms and controlling their behavior in an interpretable way (Singh et al., 2024). Steering has emerged as a common approach for this goal because it is simple, efficient, and works at inference time (Rimsky et al., 2024; Gao et al., 2025). It adds a concept-related vector to internal representations to promote or suppress a target concept in the model's outputs.

However, practical steering effectiveness is often inconsistent across behaviors and individual inputs (Tan et al., 2024). Existing accounts largely rely on the Linear Representation Hypothesis (LRH) (Park et al., 2024), which assumes that concepts are encoded linearly in the models and that independent concepts can be orthogonalized for lossless control. Based on this idealization, several methods attempt to estimate steerability using criteria such as representation separability (Braun et al., 2025), treating higher separability as indicating purer concept extraction. In practice, these estimates remain controversial and do not consistently correlate with actual steering outcomes across different settings and datasets (Bas & Novak, 2025). This suggests that lossless concept disentanglement is not achievable in realistic settings, and that steering unpredictability reflects a more fundamental underlying geometric mechanism rather than incidental noise.

Based on this finding, we propose the **Cylindrical Representation Hypothesis (CRH)**: representation differences arise from a linear combination of multiple, potentially non-orthogonal concepts, which induces a sample-specific axis-orthogonal geometry. As illustrated in Figure 1, CRH provides a principled explanation for steering instability where LRH falls short. LRH assumes a single global concept direction, implying that any two steering vectors with similar angular alignment to the target should yield comparable success. In practice, however, such vectors can still produce different outcomes. Under CRH, steering is characterized by three elements: an *axis*, a *normal plane*, and *sensitive sectors*. Even if two steering vectors are angularly close, their projections into the sample-specific normal plane may fall into different sectors. One may enter a sensitive sector that facilitates concept activation, while the other lands in a non-sensitive region that suppresses or delays it. This interaction reveals that steerability is intrinsically sample-specific and governed by the local phase within the cylindrical structure.

These components differ in predictability. When the latent concept composition of a sample is unknown, the magnitude of the normal-plane component can still be estimated from the axis-based decomposition, providing an approximate measure of steering intensity. However, the sensitive sector within the plane cannot be inferred from the same information. As a result, steering effects are highly sample-specific: overall steerability trend is observable, but the outcome for an individual sample remains unpredictable. This explains the difficulty of predicting steering behavior and the limitations of linear response assumptions.

We validate CRH through extensive verification experiments of 100 concepts, spanning multiple model architectures and steering implementations. Across all settings, a consistent cylindrical structure emerges, demonstrating that CRH robustly captures and explains the sample-specific behavior observed in practical steering.

In summary, our contributions are as follows: *(i)* We propose the Cylindrical Representation Hypothesis, which extends LRH by allowing overlapping concepts. *(ii)* We show that the cylindrical geometry explains sample-specific and irregular steerability. *(iii)* We empirically validate the cylindrical structure across diverse models and steering methods.

# 2. Limitations of the Linear Representation Hypothesis

## 2.1. Background

This section reviews the essential background needed to motivate our problem setting and clarify the context of our approach, focusing on steering methods and the Linear Representation Hypothesis. A broader discussion of related work is provided for reference in Appendix 7.

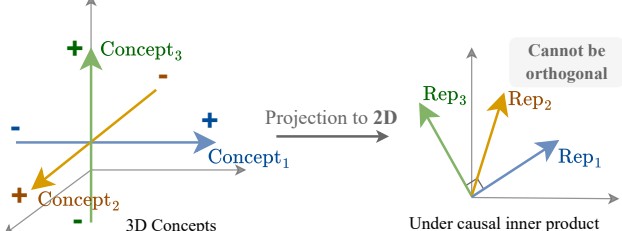

*Figure 2.* A counterexample of independent control. Three independent concepts defined in a three-dimensional latent space cannot remain orthogonal when represented in two dimensions. As a result, intervening on one concept inevitably affects others.

**Steering** modifies model outputs at inference time by adding a concept-related vector to internal representations, typically constructed from differences between positive and negative examples (Rimsky et al., 2024). Most existing steering methods rely on this linear intervention scheme and are commonly justified using LRH.

**LRH** (Park et al., 2024) assumes that concepts correspond to linear directions in representation space, so that adding a concept vector can control model behavior. It further introduces *causal separability*, which assumes that logically non-interfering concepts can be represented as orthogonal directions under a suitable "causal inner product". In this idealized setting, steering is expected to be stable and lossless. Formal definitions are deferred to Appendix B.

## 2.2. Steering Instability under LRH

In practice, steerability can be unstable. Steering outcomes vary with the construction dataset, the target concept, and individual test samples (Tan et al., 2024). Some work tries to estimate steerability in advance using linear criteria such as representation separability, where larger separation is taken to mean less overlap with other concepts (Braun et al., 2025). However, these estimates do not always correlate well with actual steering outcomes (Bas & Novak, 2025). Such approaches follow the LRH ideal settings, treating steering variability as reducible interference and interpreting higher separability as closer to independent control.

**A Counterexample for Ideal Lossless Control.** We argue that this idealization cannot be fully achieved: Even when concepts are logically independent, interference between their representations is unavoidable. In a $d$-dimensional space, at most $d$ directions can be mutually orthogonal. Once the number of independent concepts exceeds $d$, representational overlap must occur. Figure 2 illustrates this with a simple example: Consider concepts of "sign flips along the coordinate X/Y/Z" of a three-dimensional space. Because they can change independently and correspond to orthogonal directions, they are causally separable. However, when represented in a two-dimensional space, no causal inner product can guarantee mutual orthogonality.

This counterexample shows that concept overlap is not merely a result of noise, but a fundamental geometric constraint. Consequently, irregular steering behavior should not be merely viewed as an accidental failure of vector quality. Instead, it reflects an inherent limitation of the LRH idealization. This motivates a refinement of LRH to realistic settings with unavoidable interference, enabling a more faithful modeling of steering mechanisms.

# 3. Cylindrical Representation Hypothesis

To overcome the limitations of LRH and explain the sample-specific nature of steerability, we introduce the Cylindrical Representation Hypothesis (CRH), a refinement of LRH that allows for interference between concepts. It is named after the cylindrical structure that locally forms around each sample and directly shapes steering effectiveness.

## 3.1. Preliminaries

**Concepts.** We denote a concept as a directed pair of semantically describable attributes of texts that can change independently, such as male$\Rightarrow$female. This definition follows prior work where a concept is treated as an operational variable whose variation causally influences the generated output (Rimsky et al., 2024; Park et al., 2024).

**Representation Space.** To unify different steering implementations and to simplify the analysis that follows, we define an idealized *output representation space*, in which each point causally corresponds to a specific model output. Steering a point in this space, therefore, induces a corresponding change in the generated output. We conduct our subsequent analysis in this representation space.

**Inherited and Extended Assumptions from LRH.** CRH builds on LRH and inherits its basic representational assumptions. Concepts are treated as fixed linear directions in the representation space, and manipulating these directions alters the corresponding concepts in the model's output. Unlike LRH, CRH does not require logically independent concepts to be represented as orthogonal directions.

Full formulation and preliminaries are in Appendix C.

## 3.2. Derivation of the Cylindrical Geometry

**Core Assumption: Overlapping Linear Contributions.** CRH assumes that the semantic difference between two representations arises from the joint contribution of multiple concepts. Formally, assume that the output representation space contains $n$ concept directions, each represented by a unit vector $\{\mathbf{a}^{(i)}\}_{i=1}^{n} \subset \mathbb{R}^d$. For any two representations $\mathbf{r}_a, \mathbf{r}_b \in \mathbb{R}^d$, let us define the difference vector:

$$\mathbf{v}_d = \mathbf{r}_a - \mathbf{r}_b. \tag{1}$$

CRH posits that $\mathbf{v}_d$ can be expressed as a linear combination of these concept directions:

$$\mathbf{v}_d = \sum_{i=1}^{n} \mathbf{v}^{(i)} = \sum_{i=1}^{n} \alpha^{(i)} \mathbf{a}^{(i)}, \tag{2}$$

where $\alpha^{(i)} \in \mathbb{R}$ denotes the contribution of concept $i$ to the semantic difference between $\mathbf{r}_a$ and $\mathbf{r}_b$.

**Axis–Orthogonal Decomposition Induced by CRH.** Under CRH, the representation difference vector $\mathbf{v}_d$ naturally induces an axis-orthogonal decomposition of concepts, which forms the geometric basis of the later cylindrical interpretation of steering, as shown in Figure 3(a). Specifically, $\mathbf{v}_d$ defines a central axis, while all orthogonal concept components are constrained to remain balanced.

Formally, we decompose each concept direction with respect to $\mathbf{v}_d$ using standard projection:

$$\mathbf{v}^{(i)} = \langle \mathbf{v}^{(i)}, \mathbf{a}_d \rangle \mathbf{a}_d + \left( \mathbf{v}^{(i)} - \langle \mathbf{v}^{(i)}, \mathbf{a}_d \rangle \mathbf{a}_d \right). \tag{3}$$

For brevity, we write:

$$\mathbf{v}^{(i)} = d^{(i)} \mathbf{a}_d + \mathbf{v}_\perp^{(i)}, \tag{4}$$

where $\mathbf{v}_\perp^{(i)} \perp \mathbf{a}_d$. Substituting this decomposition into the CRH formulation Equation (2) yields

$$\mathbf{v}_d = \sum_{i=1}^{n} \mathbf{v}^{(i)} = \underbrace{\left( \sum_{i=1}^{n} d^{(i)} \right)}_{\|\mathbf{v}_d\|} \mathbf{a}_d + \underbrace{\sum_{i=1}^{n} \mathbf{v}_\perp^{(i)}}_{\mathbf{0}}. \tag{5}$$

As a result, a balanced state emerges in the subspace orthogonal to $\mathbf{v}_d$, where the contributions of all concept components cancel each other out.

Now, consider steering toward a target concept $c$ for an original representation $\mathbf{r} \in \mathbb{R}^d$. Let $\mathbf{r}_c$ denote a fluent output representation that expresses $c$, and define $\mathbf{v}_d = \mathbf{r}_c - \mathbf{r}$. In this case, the orthogonal balance can be written as

$$\mathbf{v}_\perp^{(c)} + \sum_{i \neq c} \mathbf{v}_\perp^{(i)} = \mathbf{0}. \tag{6}$$

Thus, steering moves $\mathbf{r}$ along $\mathbf{v}_d$ toward $\mathbf{r}_c$, where $\mathbf{r}_c$ typically corresponds to a fluent output expressing concept $c$. We can view the summation term as constraints to $\mathbf{v}_\perp^{(c)}$, ensuring the output coherence and stability.

**A Sample-Specific Normal Plane.** The orthogonal components of concepts $\{\mathbf{v}_\perp^{(i)}\}$ lie in a high-dimensional subspace, which is difficult to analyze. Therefore, we focus on a two-dimensional normal plane that summarizes the most important orthogonal variation for each sample, as illustrated in Figure 3(b). We define this normal plane $\mathcal{P}_d$ as

$$\mathcal{P}_d = \text{span}\left( \mathbf{a}_\perp^{(c)}, \text{PC}_1(\{\mathbf{a}_\perp^{(i)}\}_{i \neq c}) \right), \tag{7}$$

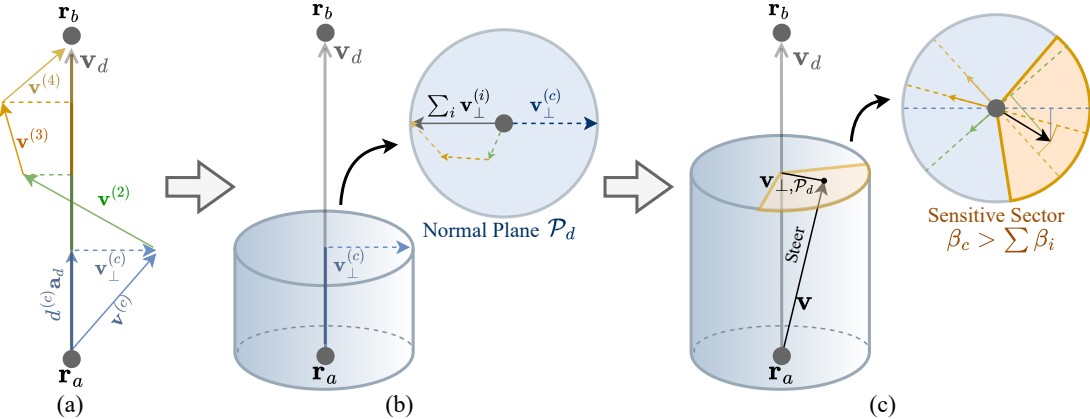

**Figure 3.** **Geometric structure induced by CRH.** (a) Each concept vector $\mathbf{v}^{(i)}$ decomposes into components parallel and orthogonal to the difference vector $\mathbf{v}_d$, and $\mathbf{v}_d$ is a weighted sum of concept vectors that defines the central axis. (b) The orthogonal components balance and form a sample-specific normal plane $\mathcal{P}_d$. (c) A steering vector splits into an axis-aligned and a normal-plane component, whose phase determines whether steering enters a sensitive or a non-sensitive sector, influencing steering effects.

where $\mathrm{PC}_1(\cdot)$ denotes the first principal direction. This choice keeps two complementary sources of variation. The first direction, $\mathbf{a}_\perp^{(c)}$, represents the full orthogonal component of the target concept. The second direction captures the dominant effects of all remaining concepts. Let $\mathrm{Proj}_{\mathcal{P}_d}(\cdot)$ denote projection onto $\mathcal{P}_d$, and define

$$\mathbf{v}_{\perp,\mathcal{P}_d}^{(i)} = \mathrm{Proj}_{\mathcal{P}_d}(\mathbf{v}_\perp^{(i)}). \quad (8)$$

Because the original orthogonal components balance each other, this balance is preserved after projection. As a result, the projected components satisfy:

$$\sum_{i=1}^{n} \mathbf{v}_{\perp,\mathcal{P}_d}^{(i)} = \mathbf{0}, \quad (9)$$

which shows that the normal plane retains the essential cancellation structure among orthogonal concept contributions.

**Phase and Sensitive Sectors in the Normal Plane.** Steering along $\mathbf{v}_d$ represents the most suitable way to induce the target concept, since this direction points toward an output state that preserves semantic coherence while expressing the target concept. In this ideal case, the axis-aligned drive alone is sufficient to activate the concept.

In practice, however, steering vectors are estimated from multiple samples and typically deviate from this ideal direction. Such deviations lie in the normal plane $\mathcal{P}_d$ induced by $\mathbf{v}_d$, and they play a critical role in determining whether the axis-aligned drive can effectively produce the target concept. We decompose a steering vector $v$ as

$$\mathbf{v} = \mathbf{v}_{\mathrm{axis}} + \mathbf{v}_{\perp,\mathcal{P}_d} + \boldsymbol{\epsilon}, \quad (10)$$

where $\mathbf{v}_{\mathrm{axis}} \parallel \mathbf{a}_d$, $\mathbf{v}_{\perp,\mathcal{P}_d} \in \mathcal{P}_d$. Here, $v_d$ captures the ideal axis-aligned component, $\mathbf{v}_{\perp,\mathcal{P}_d}$ represents the deviation within $\mathcal{P}_d$, and $\boldsymbol{\epsilon}$ denotes a residual component that is weakly related to steering behavior.

Within the normal plane $\mathcal{P}_d$, this deviation can be expressed as a combination of orthogonal concept components:

$$\mathbf{v}_{\perp,\mathcal{P}_d} = \beta_c \mathbf{v}_{\perp,\mathcal{P}_d}^{(c)} + \sum_{i \neq c} \beta_i \mathbf{v}_{\perp,\mathcal{P}_d}^{(i)}, \quad (11)$$

where $\beta_c$ denotes the relative contribution of the target concept, and $\beta_i$ denotes the relative contributions of non-target concepts. The direction of $v_{\perp,\mathcal{P}_d}$ within $\mathcal{P}_d$, referred to as its *phase*, quantifies the relative dominance between the target concept and other concepts.

When the contribution of the target concept exceeds the combined contribution of all other concepts, the plane-level deviation reinforces the axis-aligned drive and strongly promotes activation of the target concept. Conversely, when the combined influence of non-target concepts dominates, their effect outweighs that of the target concept, leading to weak activation or suppression. Based on this observation, we partition the normal plane into *sensitive sectors* using the following sufficient conditions:

high-sensitivity sector: $\quad \beta_c > \sum_{i \neq c} \beta_i, \quad (12)$

low-sensitivity sector: $\quad \beta_c \leq \sum_{i \neq c} \beta_i. \quad (13)$

**A Cylindrical Geometric Interpretation.** Together, these properties induce a cylindrical geometry around each sample, as illustrated in Figure 3. The difference vector $\mathbf{v}_d$ defines a central axis that characterizes the primary semantic transition. Orthogonal to this axis, a sample-specific normal plane $\mathcal{P}_d$ captures the residual variations arising from multiple concepts. The directions within this plane admit a phase structure, given by the orientation of $v_{\perp,\mathcal{P}_d}$ in $\mathcal{P}_d$, which partitions the plane into regions with distinct geometric roles. This axis–plane–phase configuration forms the structural basis of CRH.

## 3.3. Steering under CRH

The cylindrical model treats the axis, the normal plane, and the phase as largely sample-intrinsic geometric properties. Steering is therefore the interaction between a generic steering vector and this sample-specific geometry.

For a fixed sample and target concept, any steering vector $\mathbf{v}$ can be decomposed following Equation (10) as follows: $\mathbf{v} = \mathbf{v}_{\text{axis}} + \mathbf{v}_{\perp,\mathcal{P}_d} + \boldsymbol{\epsilon}$, where $\mathbf{v}_{\text{axis}}$ aligns with the axis, $v_{\perp,\mathcal{P}_d}$ lies in the normal plane, and $\boldsymbol{\epsilon}$ is a residual term. These components play distinct roles. The axis component $\mathbf{v}_{\text{axis}}$ drives a stable semantic shift toward a concept-expressing state. The plane component $\mathbf{v}_{\perp,\mathcal{P}_d}$ controls whether this shift successfully activates the target concept, with its phase in $\mathcal{P}_d$ determining facilitation or diversion by competing concepts. Its magnitude further modulates the rate of concept emergence, while larger values also increase the risk of semantic drift. Effective steering thus requires alignment between an axis-aligned push and a favorable phase within $\mathcal{P}_d$. Additional discussion is provided in Appendix E.

# 4. Predictability Properties under CRH

CRH introduces a sample-specific cylindrical structure that explains irregular steering behavior, which naturally raises the question of *whether steering effects are determined by observable quantities*. For steering to be truly controllable, two conditions must hold: (*i*) the magnitude of the normal-plane component must be inferable, and (*ii*) the phase structure that governs steering effectiveness must also be determined. This section investigates the determinability of these two aspects in turn, in order to clarify to what extent CRH admits predictable and controllable steering behavior.

### 4.1. Predictability of Normal-Plane Magnitude

**Theorem 4.1.** *Under CRH, the magnitude of the normal-plane component $\|\mathbf{v}_{\perp,\mathcal{P}_d}\|$ is predictable from the decomposition of the steering vector with respect to the axis $\mathbf{a}_d$.*

**Proof sketch.** The goal is to construct an observable quantity whose variation is consistent with the latent normal-plane effect. Under CRH, the concept-related variation lies within the normal plane induced by $\mathbf{v}_d$, which provides a geometric containment relation between latent and observable components.

We define the observable quantity as the magnitude of the component of $\mathbf{v}$ orthogonal to the axis $\mathbf{a}_d$, i.e., its projection onto $\mathcal{P}_d$. To compare it with the latent effect, both quantities can be written in quadratic form. The observable plane magnitude corresponds to a projection onto $\mathcal{P}_d$, while the latent concept contribution corresponds to a projection onto an unknown subspace contained within $\mathcal{P}_d$.

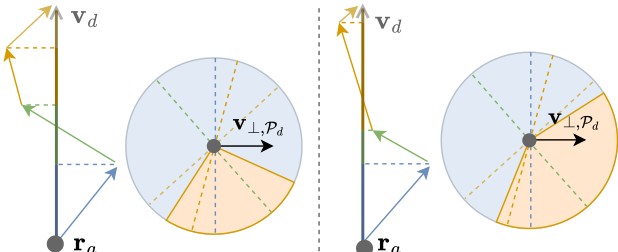

*Figure 4.* Non-predictability of steering effectiveness on normal-plane phase, where a single difference-vector direction can correspond to multiple possible concept compositions and induce different sensitive sectors.

Because of this containment, the two projections satisfy a nesting relation, which ensures that the observable projection captures all variations of the latent one. By examining their gradients, one can show that the observable quantity and the latent effect always change in aligned directions. In particular, increasing the magnitude of the observable plane component necessarily increases the corresponding latent contribution.

Therefore, the orthogonal component $\|\mathbf{v}_{\perp,\mathcal{P}_d}\|$ provides a consistent and reliable proxy for the true normal-plane effect, making this part of the steering mechanism predictable from observable quantities. Details are given in Appendix F.1.

### 4.2. Non-predictability of Sector Sensitivity

While the plane magnitude is predictable, the phase within the normal plane depends on how multiple concepts combine. This dependence leads to information loss. Figure 4 provides an intuitive illustration of this non-determinability, showing that identical normal-plane directions can correspond to opposite steering outcomes under different latent concept configurations. Full proofs are given in Appendix F.2 and F.3.

**Lemma 4.2.** *In a $d$-dimensional representation space with more than $d$ concept directions, the mapping from concept strengths to the difference vector $\mathbf{v}_d$ is non-injective.*

Lemma 4.2 implies that a single observable axis can correspond to multiple latent concept configurations.

**Theorem 4.3.** *Under CRH, the sensitive sector in the normal plane $\mathcal{P}_d$ is not reliably predictable from $\mathbf{v}_d$ and $v$.*

**Proof sketch.** The effect of a steering direction within $\mathcal{P}_d$ depends on the balance between target and non-target concept components. Under CRH, $\mathbf{v}_d$ aggregates multiple concept contributions via a non-injective mapping, so the same $\mathbf{v}_d$ can correspond to different latent configurations.

Consequently, even with fixed $\mathbf{v}_d$ and plane direction of $v$, different latent configurations can induce opposite effects, yielding counterexamples to any deterministic prediction based only on $(\mathbf{v}_d, v)$.

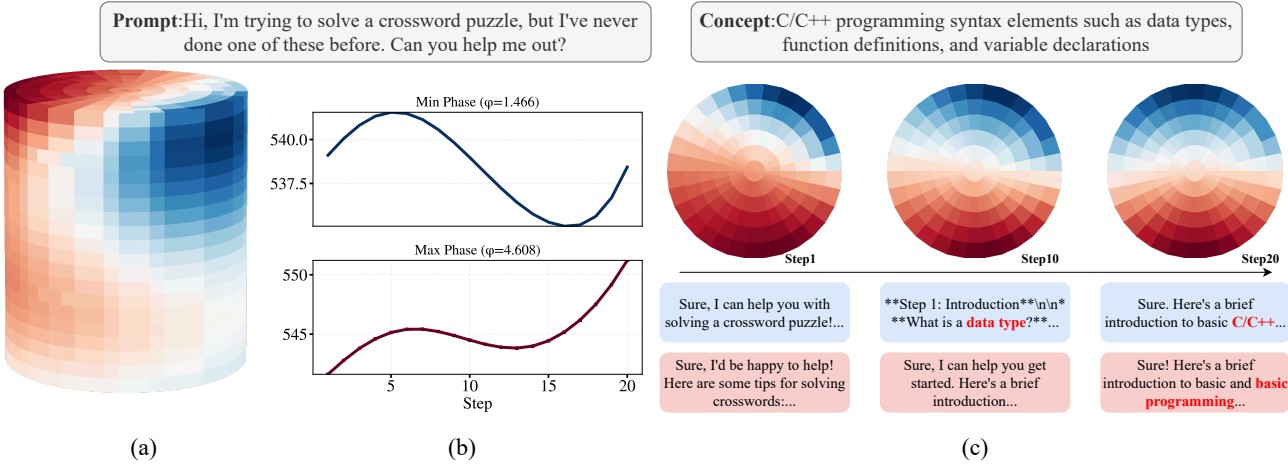

*Figure 5.* **Probed cylindrical structure of CRH for a fixed sample.** (a) We show the loss distribution over the entire cylindrical structure. (b) We plot loss trajectories along the axis for the phases with the **minimum** and **maximum** average loss. (c) We present normalized loss distributions over the normal plane at selected steering steps, showing stable sector patterns across steps. For each plane, we show the outputs corresponding to the minimum and the maximum loss regions and highlight target-concept-related fragments in **bolded red**.

Full proof of Lemma 4.2 and Theorem 4.3 are given in Appendix F.2 and F.3.

### 4.3. Implications of CRH for Steerability

CRH attributes irregular steerability to sample-specific concept composition. Different samples contain different proportions between target and non-target concepts, which leads to different effective decompositions of the same steering vector. This ratio controls the phase behavior in the normal plane and varies across samples, making single-instance outcomes hard to predict. In contrast, the normal plane depends only on which concepts are present and remains relatively stable. As a result, steerability appears irregular at the sample level, but remains measurable in aggregate. Further discussion is available in Appendix E.

## 5. Probing the Cylindrical Structure

CRH is formulated in an idealized output representation space, where the cylindrical structure is defined. To empirically test the presence of this cylindrical structure, we construct a controlled approximation of the output space using the model's internal hidden states. This section examines whether actual steering behavior in hidden representations follows the geometric patterns predicted by CRH.

### 5.1. Probing Experiment Setup

**Probing Method.** Observing the latent geometry of CRH requires a method that establishes a direct link between representations and final outputs. We use one-shot steering vector optimization (Dunefsky & Cohan, 2025) to achieve this coupling.

By freezing the model parameters and optimizing a trainable vector to maximize the probability of a target sentence while suppressing the original output, we effectively perform a reverse mapping from the output space. This setup provides a high-purity experimental window by filtering out unrelated semantic noise, which makes the structural alignment between internal steering and output objectives observable. Optimization details are provided in Appendix G.1.

**Probing Process.** To capture the local geometry around a specific sample more faithfully, we optimize steering vectors under multiple norm budgets and use the resulting vector set to construct a cylindrical coordinate system. This procedure first identifies the dominant optimization direction as the cylinder axis, then uses the next two dominant variation directions to define the normal plane. By probing different axial positions, radii, and phases in this coordinate system, we systematically map the resulting loss landscape and the corresponding generation behavior.

For each sample, we first build the difference vector $\mathbf{v}_d$ by contrasting the last-token residual-stream activations of an original prompt and its concept-constrained counterpart. We then optimize a steering vector under a sequence of norm budgets $\|\mathbf{v}\| \leq w\|\mathbf{v}_d\|$ with $w \in \{0.1, 0.2, \ldots, 2\}$, initializing each run with $\mathbf{v} = w\mathbf{v}_d$ and running 30 optimization iterations with learning rate $0.1$ to obtain a set of optimized vectors and their losses. Next, we apply Principal Component Analysis (PCA) to this vector set and use the leading principal direction as the cylinder axis, while the next two components span the normal plane. Finally, by fixing an axial coordinate and sweeping directions within the normal plane across multiple radius values and phases, we map the loss landscape and model outputs around the sample.

## 5.2. Visualization and Analysis

Figure 5 shows the results of the probing experiments and highlights several patterns predicted by CRH.

Figure 5(a) visualizes the loss distribution over the full cylindrical structure, revealing clear phase-dependent variation in the normal plane and pronounced differences between opposite regions of the cylinder.

Figure 5(b) plots the loss trajectories along the axis for the phases with the lowest and the highest average loss. In this example, the low-loss phase shows a consistent decrease in the loss as steering progresses, while the high-loss phase exhibits an overall increase, demonstrating sustained promotion and suppression of the target concept, respectively.

Figure 5(c) shows the normalized loss distributions over the normal plane at different steering steps. The locations of low- and high-loss regions remain largely unchanged, indicating stable sensitive and non-sensitive sectors. The outputs corresponding to these regions show that the target concept emerges earlier in sensitive sectors and much later in non-sensitive ones.

Across samples, sector patterns vary in form, but consistently support the cylindrical structure and phase-dependent sensitivity predicted by CRH. We further provide more cases in Appendix G.2.

Furthermore, Appendix K.1 shows that the top three PCA components can capture most variance of the optimized vectors, and Appendix K.2 proves that the structured loss landscape is not random.

## 6. Verification via Observable Implications

CRH explains why steering can be unstable, but this relies on one core assumption: a sample-level cylindrical geometry exists in the model's representation space. Since the true concept directions are unknown, this structure is not directly observable. We therefore derive observable consequences of the hypothesis and test them empirically.

### 6.1. Observable Implications of the Cylindrical Model

Here, we present three observable implications from the cylindrical model. The main goal is to turn the geometric assumptions into measurable trends verifiable in experiments.

**Implication 1: Existence of steering effect decomposition.** Based on the decomposition $\mathbf{v} = \mathbf{v}_d + \mathbf{v}_{\perp,\mathcal{P}_d} + \boldsymbol{\epsilon}$ in Section 3.3, the steering vector contains two main components with distinct effects on the output. The axis component $\mathbf{v}_d$ drives a stable semantic shift toward the target concept. In contrast, the plane component $\mathbf{v}_{\perp,\mathcal{P}_d}$ amplifies the strength of concept expression, but reduces output stability.

Therefore, increasing $\|\mathbf{v}_{\perp,\mathcal{P}_d}\|$ while keeping $\|\mathbf{v}_d\|$ at a similar scale would produce two systematic effects: (1) **Faster concept activation:** The target concept $c$ emerges at smaller overall steering magnitudes, due to stronger influence within the normal plane. (2) **Earlier loss of coherence:** The output becomes unstable or semantically incoherent more easily, as the orthogonal component pushes the representation away from a stable semantic trajectory.

**Implication 2: Correlation signature of plane determinability.** The next question is whether the normal plane is predictable from the axis. Assume that the normal plane is determined by $\mathbf{v}_d$. Then the effect of a steering vector depends on how it splits into axis-aligned push and plane deviation, which can be summarized by the angle $\theta$ between $\mathbf{v}$ and $\mathbf{v}_d$. We show in appendix F.4 that, under this assumption of determinability, that the steerability of a certain concept $c$ can be written in the following form:

$$\mathrm{St}_c(\mathbf{r}; \mathbf{v}) \propto \|\mathbf{v}_d\|^{m+n} \sin^m \theta \cos^n \theta, \tag{14}$$

for fixed $m > 0, n > 0$ shared across samples.

Let $k = m + n$. Equation (14) implies that after removing the scale $\|\mathbf{v}_d\|^k$, the remaining variation should be explained by a single mixed-power term in $\sin \theta$ and $\cos \theta$. Concretely, for a fixed $k$, scanning $m \in (0, k)$ yields a correlation:

$$\frac{\mathrm{St}_c(\mathbf{r}; \mathbf{v})}{\|\mathbf{v}_d\|^k} \propto \sin^m \theta \cos^{k-m} \theta, \tag{15}$$

and the model predicts a single clear maximum at the correct split $m$. If the plane is not determined by $\mathbf{v}_d$, this peak can become weak, unstable across settings, or non-unique.

**Implication 3: Sector non-determinability breaks similarity transfer.** Finally, we ask whether the *most favorable phase* inside the normal plane is also determined by $\mathbf{v}_d$. Let $\Phi_c(r)$ denote the sensitive sector of sample $r$ for concept $c$, i.e., the set of plane phases where steering favors concept $c$. If $\Phi_c(r)$ were determined by $\mathbf{v}_d$ through a simple rule, similar difference vectors would imply similar sectors:

$$\mathbf{v}_d(\mathbf{r}_1, c) \approx \mathbf{v}_d(\mathbf{r}_2, c) \ \Rightarrow \ \Phi_c(\mathbf{r}_1) \approx \Phi_c(\mathbf{r}_2). \tag{16}$$

Thus, under the same steering vector $\mathbf{v}$, the two samples should show similar steering patterns, such as similar concept-emergence time and similar success or failure. If the sector is not determined by $\mathbf{v}_d$, then $\mathbf{v}_d$ similarity alone will not reliably transfer to outcomes: two samples with similar $\mathbf{v}_d$ can still fall into different sectors and exhibit very different steering behaviors under the same $\mathbf{v}$.

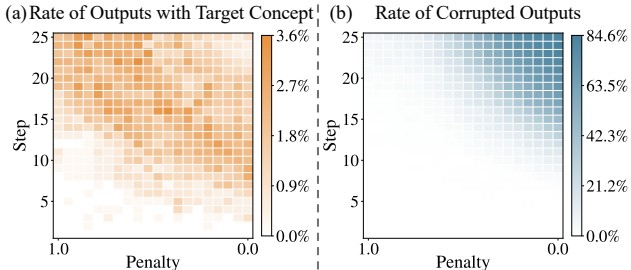

*Figure 6.* Effect of penalizing the normal-plane component on steering outcomes: (a) target concept activation and (b) output corruption, illustrating the trade-off predicted by CRH. Shown are results for layer 9 in Gemma-2B-IT.

## 6.2. Experimental Validation

### 6.2.1. EXPERIMENTAL SETUP

**Models and intervention layers.** We use two models with different scales and architectures: Gemma-2B-IT (Gemma Team et al., 2024) and LLaMA2-7B-Chat (Touvron et al., 2023). Following the protocol in AxBench (Wu et al., 2025), we select two representative layers for each model, located at roughly one-third and two-thirds of the network depth. These correspond to layers 9 and 13 for Gemma, and layers 16 and 24 for LLaMA2. All interventions are applied to the residual stream at the selected layers.

**Datasets.** We construct a concept-level benchmark using concepts collected by AxBench (Wu et al., 2025), together with prompts from AlpacaEval (Li et al., 2023b). The selected concepts are uniformly sampled across text, code, and math domains. Following the AxBench procedure, training pairs consist of "negative" examples from raw model outputs and "positive" examples generated via concept-specific instruction augmentation. Detailed dataset statistics and construction details are provided in Appendix H.1.

**Steering Implementation.** We construct the steering vectors using several standard methods, including DiffMean (Rimsky et al., 2024), PCA-based steering (Zou et al., 2023), Mean-Centering (MC) (Jorgensen et al., 2023), and probe-based steering (Li et al., 2023a); we also test different intervention layers. During inference, we apply steering as $\mathbf{r} \leftarrow \mathbf{r} + \lambda \mathbf{v}$, and sweep the steering strength $\lambda$ over a predefined range to examine how model behavior changes. In the main text, we primarily present results for DiffMean applied uniformly to all prompt tokens; Appendix H.2 compares various methods and intervention settings.

**Steerability Evaluation.** We evaluate the target concept expression and output coherence using LLM-as-a-judge; see Appendix D for detail. We quantify steerability by measuring how quickly the target concept $c$ appears as steering strength increases. For a given steering vector $v$, the proportion of successful samples $p(\lambda)$ is computed at each value of $\lambda$. A linear model $p(\lambda) \approx a\lambda + b$ is then fitted.

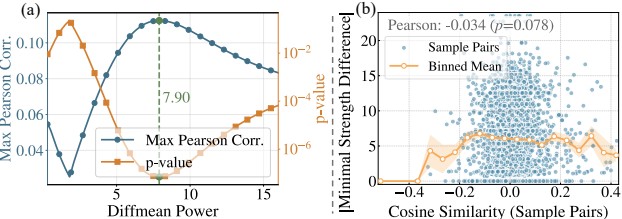

*Figure 7.* Verification of CRH determinability. (a) The maximum correlation exhibits a single peak as the total exponent increases, with the minimum p-value attained near the peak. (b) The difference-vector similarity does not correlate with steerability similarity for a fixed concept (Gemma-2B-IT, layer 9).

We define the steerability score as

$$\text{St}_c(\mathbf{v}) = \frac{a}{\|\mathbf{v}\|}, \tag{17}$$

which represents the increase in the success rate per unit norm of the steering vector.

In our experiments, we compute the reference difference vectors $\mathbf{v}_d$ for test samples to define sample-specific axes in the CRH analysis. These reference vectors are used only for geometric interpretation and are not involved in steering vector construction.

### 6.2.2. EXPERIMENTAL RESULTS

**Validation of Implication 1.** In this experiment, we use a penalty to control the orthogonal component of a steering vector. For each test sample, the method decomposes the steering vector $\mathbf{v}$ into $\mathbf{v} = \mathbf{v}_{\text{axis}} + \mathbf{v}_\perp$, where $v_d$ aligns with the sample-specific difference vector and $v_\perp$ lies in the orthogonal subspace. The projection process onto $\mathcal{P}_d$ remains fixed for each sample. As a result, changing $\|\mathbf{v}_\perp\|$ directly changes $\|\mathbf{v}_{\perp,\mathcal{P}_d}\|$.

The method applies the following penalty:

$$\mathbf{v}_\perp \leftarrow (1 - \rho)\,\mathbf{v}_\perp, \tag{18}$$

with $\rho \in [0, 1]$. Larger $\rho$ reduces the orthogonal contribution. Setting $\rho$ to 1 removes this contribution completely. Our experiment sweeps $\rho$ from 0 to 1 in 25 steps. For each value of $\rho$, the steering strength $\lambda$ is swept over a range that depends on the specific steering configuration (see Appendix H.2), and we uniformly divide this range into 25 steps. For each pair $(\rho, \lambda)$, the method records whether the output expresses the target concept or becomes invalid.

Figure 6 shows the results for Gemma at layer 9. Two clear trends emerge: As shown in Figure 6(a), smaller $\rho$ leads to earlier concept emergence at lower $\lambda$. In Figure 6(b), smaller $\rho$ also causes invalid outputs to appear earlier. When $\rho = 1$, the outputs remain the most stable. These trends closely match the predictions of the cylindrical model. See Appendix J.1 for the full results.

**Validation of Implication 2.** Following the implication in Section 6.1, the experiment tests whether the normal plane can be predicted from the axis direction $\mathbf{v}_d$. The analysis varies the total exponent $k$ and, for each $k$, evaluates how well the normalized steerability follows a mixed-power function of the angle $\theta$. This test checks the correlation pattern implied by Equation (14). Specifically, for each $k$, we define the maximum achievable rank correlation as:

$$\rho_k \;=\; \max_{m \in (0,k)} \;\rho\!\left(\tfrac{\mathrm{St}_c(\mathbf{v})}{\|\mathbf{v}\|^k},\; \sin^m \theta \, \cos^{k-m} \theta\right), \quad (19)$$

where $\rho(\cdot,\cdot)$ denotes the Pearson rank correlation across concepts. The corresponding $p$-value is recorded for each $k$.

Figure 7(a) shows the results for Gemma at layer 9. The correlation curves exhibit a clear unimodal structure as $k$ varies. The peak Pearson correlation coincides with the minimum $p$-value, indicating strong statistical significance. These results follow the theoretical prediction and support the claim that the normal plane is determined by $\mathbf{v}_d$. Detailed experimental settings are provided in Appendix J.2.

**Validation of Implication 3.** To test whether sensitive sectors are determined by difference-vector similarity, we analyze the relationship between steering patterns and sample similarity. For each concept, we record the steering strength at which the target concept first appears. For any pair of test samples, we compute the cosine similarity between their difference vectors and the absolute difference in their respective steering strengths. We then aggregate these pairs across all concepts. Figure 7(b) shows the resulting distribution. We find no meaningful correlation between difference-vector similarity and steering behavior (Pearson $= -0.034$, $p > 0.05$), i.e., sample similarity does not predict steering success or pattern similarity, which supports the non-determinability of sectors. The detailed results are given in Appendix J.3. To test whether CRH holds as model size increases, we also verify a larger LLM in Appendix K.3.

## 7. Related Work

**Activation Steering.** Activation steering controls model behavior by intervening on internal representations at inference time (Zou et al., 2023). Most methods derive steering directions from representation differences between positive and negative examples (Rimsky et al., 2024), offering an interpretable and efficient alternative to fine-tuning (Marks & Tegmark, 2024; Jiang et al., 2025). However, steering effectiveness is often sample-dependent (Tan et al., 2024), and learned directions may contain spurious or entangled features (Brumley et al., 2024). Prior work addresses these issues through vector denoising (Zhao et al., 2026), optimization-based steering (Dunefsky & Cohan, 2025; Cao et al., 2024), non-linear interventions (Vu & Nguyen, 2025; Oozeer et al., 2025), and adaptive steering mechanisms (Lee et al., 2025; Zhang et al., 2024).

Unlike these methods, which aim to improve steering performance, we seek to explain why steering remains intrinsically sample-dependent. CRH provides a geometric account of steering variability by characterizing the sample-specific structure underlying steering interventions.

**Geometric Concept Representations.** The dominant theoretical framework for concept representations is the Linear Representation Hypothesis (LRH), which models concepts as linear directions in representation space and motivates steering through vector arithmetic (Zou et al., 2023; Gurnee & Tegmark, 2024; Park et al., 2024). Subsequent work has shown that learned representations exhibit feature superposition (Elhage et al., 2022) and richer geometric structures, including circular (Engels et al., 2025), clock-like (Nanda et al., 2023; Zhong et al., 2023), and manifold-based organizations (Modell et al., 2025). Other extensions broaden LRH from token-level concepts to sentence-level or multi-token representations (Nguyen & Leng, 2025; Valois et al., 2025). Unlike these studies, which study how concepts are represented, we focus on how representation geometry shapes steering behavior. In particular, CRH relaxes LRH's orthogonality assumption while preserving linear concept representations, yielding a sample-specific cylindrical geometry that explains steering instability and the limited predictability of steering outcomes.

## 8. Conclusion and Future Work

We introduced the Cylindrical Representation Hypothesis (CRH), a geometric extension of the Linear Representation Hypothesis that explains the sample-specific nature of language model steering. By modeling concept representations through a central axis, a normal plane, and sensitive sectors, CRH provides a principled account of why steering outcomes can vary substantially across samples despite similar intervention directions. Our theoretical analysis identifies which aspects of steering are predictable and which are fundamentally uncertain, while extensive experiments provide evidence for the proposed cylindrical structure.

Our results suggest that steering variability arises from intrinsic representation geometry rather than solely from imperfect steering vectors. This perspective shifts attention from finding better steering directions to understanding the geometric constraints that govern steerability. We hope CRH serves as a foundation for future work on more reliable, interpretable, and controllable representation-based interventions in large language models.

In future work, we plan to incorporate sample-specific geometry into steering algorithms toward more reliable model control. Beyond steering, these geometric principles may help understand broader phenomena in representation engineering, interpretability, alignment, and behavior control.

## Impact Statement

This work advances the understanding of how Lareg Language Models (LLMs) represent and manipulate concepts, specifically challenging the idealized Linear Representation Hypothesis. By characterizing the sample-specific cylindrical geometry of representations, we provide a mechanistic explanation for why model steering often fails on specific inputs. This contribution is crucial for the reliable deployment of LLMs across diverse applications ranging from safety alignment and truthfulness to personalization and precise attribute control. Relying on unstable steering methods in these areas can lead to unpredictable behaviors or experiences. By highlighting the intrinsic uncertainty in the "sensitive sectors" of representation space, our work encourages the community to move beyond linear assumptions and design more robust steering techniques that account for these geometric constraints.

While improved understanding of steering may ultimately enable more effective model control, the results presented here do not directly increase a model's capabilities. Instead, they help clarify the opportunities and limitations of representation-based interventions, contributing to the broader goal of building safer and more reliable AI systems.

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

## A. Limitations

While CRH offers a new geometric perspective for interpreting steering behavior, several limitations should be noted. First, CRH is proposed as a conceptual framework rather than a directly verifiable property of model representations. It assumes that sample-level difference vectors $\mathbf{v}_d$ can be expressed as combinations of underlying concept directions in an idealized output representation space. This assumption may benefit from further theoretical analysis or targeted mechanistic studies. Second, although CRH attributes steering variability to sample-specific geometric structure, our analysis primarily focuses on vector-level interactions and does not explicitly examine how fine-grained activation dynamics within individual samples give rise to such geometry. Incorporating more detailed activation-level analyzes may provide additional insight into the origins of sample-specific behavior. Finally, the empirical evaluation considers a finite set of concepts drawn mainly from text, code, and math domains, and does not examine more complex settings such as multi-step mathematical derivations, long-horizon reasoning, or agent-style tool use. These more involved scenarios may exhibit additional structure not captured in the current analysis.

## B. Further Details of LRH

**Basic Formulation.** The Linear Representation Hypothesis (LRH) assumes that language models encode semantic concepts as linear directions in high-dimensional representation spaces (Zou et al., 2023; Gurnee & Tegmark, 2024). Concept strength is reflected by projection magnitude onto the corresponding direction, enabling both detection and manipulation through vector arithmetic. This linear assumption forms the core representational basis of most steering methods.

**Causal Separability and Causal Inner Product.** In addition to linearity, LRH introduces *causal separability* (Park et al., 2024), a concept-level assumption stating that logically non-interfering concepts admit independent interventions. LRH further posits the existence of a *causal inner product* under which such concepts correspond to orthogonal directions in representation space, with orthogonality reflecting the absence of causal interference.

**Implications for Steering.** Together, these assumptions provide theoretical support for steering. Linearity explains why adding a concept direction can influence model outputs, while causal separability motivates an ideal setting in which steering can be lossless and safe, as orthogonal concept directions prevent unintended interference.

## C. Further Details in CRH

To align with practical steering settings, CRH adopts the core linear representation assumption of LRH while relaxing several idealized conditions that are unlikely to hold in real models.

**Concepts.** Following Park et al. (2024), a concept is defined as a latent variable that is caused by the context $X$ and causally affects the output $Y$. We focus on binary concepts and fix an ordering for each concept (e.g., `male`$\Rightarrow$`female`) so that the sign of a representation is well-defined.

**Internal Representations.** LRH distinguishes embedding-side representations for concept detection and unembedding-side representations for concept manipulation. In practice, steering methods often intervene at intermediate representations without an explicit separation between detection and manipulation (Rimsky et al., 2024). CRH therefore treats all such feature spaces uniformly as *internal representations* of the model.

**Representation Granularity.** While the original LRH formulation is stated at the level of a single token (Park et al., 2024), practical steering commonly operates over multiple tokens and still relies on linear concept representations (Rimsky et al., 2024). CRH retains this assumption without restricting the number of tokens involved.

**Steering-Aligned Representation Space.** Steering methods differ in intervention location and scope, such as intervening on different layers or tokens (Zhang et al., 2026). To abstract away these implementation details, CRH defines an *output representation space* in which each point causally corresponds to a model output under a fixed intervention scheme. This space is aligned with steering: movements correspond to output changes, and concepts remain linearly represented.

**Relaxed Causal Separability.** Unlike LRH, CRH does not assume that logically non-interfering concepts correspond to orthogonal directions under a causal inner product. In the intervention space, concept representations may overlap even when concepts are causally separable at the semantic level. This relaxation reflects practical constraints and allows CRH to model interference between concepts.

**Rationale for the Use of Unnormalized Concept Vectors for Sector Definition.** In Section 3.2, when defining sectors, the plane component $\mathbf{v}_{\perp,\mathcal{P}_d}$ is expanded using projected concept vectors $\mathbf{v}_{\perp,\mathcal{P}_d}^{(i)}$ rather than normalized directions. This choice reflects that steering effects depend on the effective magnitude of concept contributions. Identical angular weights can lead to very different outcomes when concept vectors differ in norm. Using unnormalized vectors allows the coefficients $\beta_i$ to represent relative effective influence, whereas normalization would implicitly assume comparable effect scales across concepts, which does not hold under CRH.

## D. Steering Effect Evaluation

As LLMs demonstrate strong capabilities in data annotation (Zheng et al., 2023), we employ Llama-3.3-70B-Turbo (Grattafiori et al., 2024) as an automatic LLM-judge to classify model outputs under steering. Specifically, the judge is instructed to review the input question, the target concept, and the model response, and assign one of three labels: (i) *Normal*: a standard answer that does not reflect the target concept; (ii) *Target Concept-related*: an answer that explicitly contains or relates to the target concept; (iii) *Corrupted*: a corrupted or nonsensical output. Prompt used for evaluation can be found in the Appendix I.

To mitigate potential bias in LLM-based evaluation, we further conduct a human validation study to verify whether the LLM-assigned labels align with human judgment on the presence of the target concept. We construct the evaluation set in two stages: (1) for each concept, we randomly sample one example to ensure full concept coverage; (2) we add additional examples via label-stratified sampling to ensure sufficient representation for each predicted category.

Subsequently, we recruit a human annotator to independently review the sampled items. To ensure consistency, we instruct the annotator to follow the exact same evaluation protocol and label definitions as the LLM judge described above.

*Table 1.* Human validation of LLM-based evaluation labels. Human annotations are treated as ground truth.

| Class | Precision | Recall | F1 | Item Num. |
|---|---|---|---|---|
| Normal (non-target) | 0.84 | 0.91 | 0.88 | 25 |
| Target Concept-related | 0.96 | 0.92 | 0.94 | 26 |
| Corrupted | 0.98 | 0.96 | 0.97 | 51 |
| Weighted Avg. | 0.94 | 0.94 | 0.94 | 102 |

We report the agreement between LLM labels and human annotations using accuracy and macro-F1, as well as per-class precision, recall, and F1, as shown in Table 1. Notably, the LLM judge achieves an overall accuracy of 94%, demonstrating its reliability in classifying model outputs under steering.

## E. Steering Effects Discussion

From the CRH perspective, commonly used linear criteria can be reinterpreted geometrically. Directional agreement effectively measures the consistency of the axis-aligned projection across training samples. When training difference vectors are well aligned, the resulting steering vector is more likely to align with the sample-specific central axis at test time. This alignment stabilizes the axis component during steering and makes it easier to reliably induce the target concept.

In contrast, data separability mainly reflects the overall projection magnitude required to distinguish positive and negative samples. While this captures the strength of the axis-aligned component, it provides no information about the magnitude or phase of the normal-plane component. As a result, data separability cannot indicate whether the orthogonal contribution will facilitate or suppress concept activation. This limitation leads to weaker explanatory power compared to directional agreement, and prior work has observed cases where data separability shows little correlation with actual steering outcomes.

## F. Proofs and Derivations

### F.1. Proof of Theorem 4.1

*Proof.* We begin by stating the goal and conclusion of the proof. The goal is to show that there exists an observable function $g(\mathbf{v})$, free of unknown quantities, whose variation with respect to $\mathbf{v}$ is consistent with that of the latent effect $f(\mathbf{v})$. If such consistency holds, $g$ can be used to predict the behavior of $f$. The conclusion is that choosing $g(\mathbf{v})$ as the orthogonal component of $\mathbf{v}$ relative to $\mathbf{v}_d$ is sufficient for this purpose.

We define the observable quantity directly as the magnitude of the orthogonal component of $\mathbf{v}$ with respect to $\mathbf{v}_d$, whose unit vector is $\mathbf{a}_d$:

$$\mathbf{v}_\perp = \mathbf{v} - \langle \mathbf{v}, \mathbf{a}_d \rangle \mathbf{a}_d, \qquad g(\mathbf{v}) = \|\mathbf{v}_\perp\|. \tag{20}$$

This quantity depends only on $\mathbf{v}$ and $\mathbf{v}_d$ and is therefore fully observable.

To compare the variation trends of $g$ and the latent effect $f$, it is convenient to consider their squared magnitudes. The squared norm of the orthogonal component can be written in matrix form. Let

$$\mathbf{Q} = \mathbf{I} - \mathbf{a}_d \mathbf{a}_d^\top, \tag{21}$$

which represents the projection onto the normal plane $\mathcal{P}_d$. Then

$$\mathbf{v}_\perp = \mathbf{Q}\mathbf{v}, \qquad g^2(\mathbf{v}) = \|\mathbf{v}_\perp\|^2 = \mathbf{v}^\top \mathbf{Q}\mathbf{v}. \tag{22}$$

The latent steering effect is defined analogously as the squared projection onto the hidden concept subspace,

$$f^2(\mathbf{v}) = \|\text{Proj}_{P_d}(\mathbf{v})\|^2 = \mathbf{v}^\top \mathbf{P}\mathbf{v}, \tag{23}$$

where $\mathbf{P}$ is the corresponding projection matrix.

Both $f^2$ and $g^2$ are quadratic functions of $\mathbf{v}$. Their gradients with respect to $\mathbf{v}$ are

$$\nabla_\mathbf{v} f^2 = 2\mathbf{P}\mathbf{v}, \tag{24}$$
$$\nabla_\mathbf{v} g^2 = 2\mathbf{Q}\mathbf{v}. \tag{25}$$

Since the hidden concept subspace lies within the normal plane, the projection operators satisfy

$$\mathbf{Q}\mathbf{P} = \mathbf{P}. \tag{26}$$

Using this relation, the inner product of the two gradients is

$$\langle \nabla_\mathbf{v} f^2, \nabla_\mathbf{v} g^2 \rangle = (2\mathbf{P}\mathbf{v})^\top (2\mathbf{Q}\mathbf{v}) \tag{27}$$
$$= 4\,\mathbf{v}^\top \mathbf{P}\mathbf{Q}\mathbf{v} \tag{28}$$
$$= 4\,\mathbf{v}^\top \mathbf{P}\mathbf{v} = 4f^2(\mathbf{v}) \geq 0. \tag{29}$$

The non-negativity of this inner product shows that the variation directions of $g^2$ and $f^2$ with respect to $\mathbf{v}$ are always aligned. Therefore, the observable orthogonal component $g(\mathbf{v})$ can serve as a reliable surrogate for the latent normal-plane effect.

$\square$

### F.2. Proof of Lemma 4.2

*Proof.* Assume that we have concepts more than the dimension of the latent space. Let $\mathbf{A} : \mathbb{R}^n \to \mathbb{R}^d$ be a linear operator whose columns are the concept direction vectors $\{\mathbf{a}^{(i)}\}$. The observable difference vector $\mathbf{v}_d$ is generated by

$$\mathbf{A}\boldsymbol{\alpha} = \mathbf{v}_d, \tag{30}$$

where $\boldsymbol{\alpha} \in \mathbb{R}^n$ denotes the latent concept coefficients.

Since $\mathbf{A}$ maps an $n$-dimensional coefficient space into a $d$-dimensional observation space with $n > d$, its columns cannot be linearly independent. Consequently, there exist non-zero coefficient vectors that are mapped to zero, and the null space of $\mathbf{A}$ is non-trivial. Formally, the rank of $\mathbf{A}$ satisfies

$$\text{rank}(\mathbf{A}) \leq d, \tag{31}$$

which implies

$$\dim(\ker(\mathbf{A})) = n - \text{rank}(\mathbf{A}) \geq n - d \geq 1. \tag{32}$$

If $\boldsymbol{\alpha}_0$ is one solution to $\mathbf{A}\boldsymbol{\alpha} = \mathbf{v}_d$, then for any $\boldsymbol{\gamma} \in \ker(\mathbf{A})$,

$$\mathbf{A}(\boldsymbol{\alpha}_0 + \boldsymbol{\gamma}) = \mathbf{v}_d \tag{33}$$

also holds. Hence, different latent coefficient vectors can produce the same observable difference vector.

Therefore, the mapping from latent concept strengths to the observable representation is many-to-one, and information about the latent composition is necessarily collapsed. □

### F.3. Proof of Theorem 4.3

*Proof.* Similar to the proof above, the goal is to show that there does not exist a deterministic observable function $g(\mathbf{v}, \mathbf{v}_d)$ whose output is consistent with the true relative steering effect. Equivalently, the sign of the actual steering effect cannot be predicted from the observable quantities alone and is therefore unobservable.

Let $\mathbf{v}_d$ be the observable difference vector defining the normal plane $\mathcal{P}_d$. Let $\mathbf{v}_\perp \in \mathcal{P}_d$ denote the projection of the steering vector $\mathbf{v}$ onto this plane, and let its direction within the plane be parameterized by a phase $\phi$. For each concept $i$, the induced coefficient can be written as

$$\beta_i(\phi) = \|\mathbf{v}^{(i)}_{\perp,\mathcal{P}_d}\| \, \|\mathbf{v}_\perp\| \cos(\phi - \delta_i), \tag{34}$$

where $\|\mathbf{v}^{(i)}_{\perp,\mathcal{P}_d}\|$ denotes the magnitude of the concept direction projected onto the normal plane, and $\delta_i$ is a fixed but unobservable phase offset.

The net steering effect is defined as the target contribution minus the aggregate non-target interference,

$$f(\phi) = \|\mathbf{v}_\perp\| \Big( \|\mathbf{v}^{(c)}_{\perp,\mathcal{P}_d}\| \cos(\phi - \delta_c) - \sum_{i \neq c} \|\mathbf{v}^{(i)}_{\perp,\mathcal{P}_d}\| \cos(\phi - \delta_i) \Big). \tag{35}$$

By trigonometric superposition, the non-target term can be rewritten as a single sinusoid,

$$\sum_{i \neq c} \|\mathbf{v}^{(i)}_{\perp,\mathcal{P}_d}\| \cos(\phi - \delta_i) = B \cos(\phi - \delta), \tag{36}$$

where the amplitude $B$ and phase $\delta$ depend only on the latent non-target configuration. Thus,

$$f(\phi) = \|\mathbf{v}_\perp\| \Big( \|\mathbf{v}^{(c)}_{\perp,\mathcal{P}_d}\| \cos(\phi - \delta_c) - B \cos(\phi - \delta) \Big). \tag{37}$$

Assume, for contradiction, that there exists a deterministic observable function $g(\mathbf{v}, \mathbf{v}_d)$ such that

$$\operatorname{sgn} g(\mathbf{v}, \mathbf{v}_d) = \operatorname{sgn} f(\phi) \quad \text{for all latent configurations.} \tag{38}$$

For fixed $\mathbf{v}$ and $\mathbf{v}_d$, the value of $g(\mathbf{v}, \mathbf{v}_d)$ is uniquely determined.

However, by Lemma 4.2, the mapping from latent concept configurations to the observable difference vector $\mathbf{v}_d$ is non-injective. Therefore, even when $(\mathbf{v}, \mathbf{v}_d)$ are fixed, distinct latent configurations can induce different interference amplitudes $B$. As a result, one can choose two latent configurations that yield identical observable inputs $(\mathbf{v}, \mathbf{v}_d)$ but satisfy

$$B < \|\mathbf{v}^{(c)}_{\perp,\mathcal{P}_d}\| \quad \Rightarrow \quad f(\phi) > 0, \tag{39}$$

and

$$B > \|\mathbf{v}^{(c)}_{\perp,\mathcal{P}_d}\| \quad \Rightarrow \quad f(\phi) < 0. \tag{40}$$

Since $g(\mathbf{v}, \mathbf{v}_d)$ must take the same value for identical observables, it cannot match the sign of $f$ in both cases. This contradiction shows that no such deterministic function $g$ can exist.

Therefore, the sensitive sector $\Phi_c$, determined by the sign of the true steering effect $f$, is not a deterministic function of the observable difference vector $\mathbf{v}_d$ and is fundamentally unobservable. □

## F.4. Derivation of Equation 14

Steerability is defined as the effective concept activation induced per unit norm of the steering vector $\mathbf{v}$. Under the Cylindrical Representation Hypothesis, a successful steering intervention requires simultaneous alignment along the sample-specific axis $\mathbf{v}_d$ and deviation within its normal plane.

Let $\theta$ denote the angle between $\mathbf{v}$ and $\mathbf{v}_d$. The steering vector is decomposed as

$$\mathbf{v} = (\|\mathbf{v}\| \cos\theta)\, \hat{\mathbf{v}}_d + (\|\mathbf{v}\| \sin\theta)\, \hat{\mathbf{v}}_\perp, \tag{41}$$

where $\hat{\mathbf{v}}_d$ is the unit axis direction and $\hat{\mathbf{v}}_\perp$ lies in the normal plane $\mathcal{P}_d$.

The intrinsic semantic scale of the sample is characterized by the magnitude $\|\mathbf{v}_d\|$. The axial contribution to steering efficiency is assumed proportional to the projection of this scale onto the steering direction,

$$E_{\text{axial}} \propto \|\mathbf{v}_d\| \cos\theta. \tag{42}$$

The planar contribution reflects the orthogonal deviation required to activate the target concept,

$$E_{\text{planar}} \propto \|\mathbf{v}_d\| \sin\theta. \tag{43}$$

Assuming multiplicative interaction between axial transport and planar activation, steerability of concept $c$ is modeled as a joint power-law,

$$\text{St}_c(\mathbf{r}; \mathbf{v}) \propto (E_{\text{axial}})^n (E_{\text{planar}})^m. \tag{44}$$

Substituting the above expressions yields

$$\text{St}_c(\mathbf{r}; \mathbf{v}) \propto (\|\mathbf{v}_d\| \cos\theta)^n (\|\mathbf{v}_d\| \sin\theta)^m. \tag{45}$$

Rearranging terms gives

$$\text{St}_c(\mathbf{r}; \mathbf{v}) \propto \|\mathbf{v}_d\|^{m+n} \sin^m\theta \cos^n\theta. \tag{46}$$

The exponents $m, n$ are treated as concept-level constants. This mixed power-law defines a characteristic correlation pattern: if the normal plane is determined by $\mathbf{v}_d$, normalized steerability exhibits a unimodal peak when evaluated against $\sin^m\theta \cos^n\theta$.

# G. Further Details in Probing Experiments

## G.1. Probing Experiments Setup

**Optimization Objective.** The probing procedure follows the steering optimization framework in (Dunefsky & Cohan, 2025). We adopt the mixed steering objective, which simultaneously promotes a target output and suppresses the original model output. Let the input prompt be $x = (x_1, \ldots, x_n)$, the target output sequence be $y = (y_1, \ldots, y_m)$, and the steering vector be $v$. Denote by $P_{\text{model}}(y \mid x; v)$ the probability assigned by the model under intervention $v$. The promotion and suppression losses are defined as

$$\mathcal{L}_+(x, y; v) = -\sum_{k=0}^{m-1} \log P_{\text{model}}(y_{k+1} \mid y_{\leq k}, x; v), \tag{47}$$

$$\mathcal{L}_-(x, y; v) = -\sum_{k=0}^{m-1} \log(1 - P_{\text{model}}(y_{k+1} \mid y_{\leq k}, x; v)). \tag{48}$$

The mixed steering objective minimizes their sum,

$$\mathcal{L}_{\text{mix}}(x, y; v) = \mathcal{L}_+(x, y; v) + \mathcal{L}_-(x, y; v), \tag{49}$$

which encourages the target sequence while discouraging the original output, following the formulation in the referenced work.

**Optimization Procedure.**    For each test sample, we optimize steering vectors under multiple norm constraints, ranging from $0.1\|\mathbf{v}_d\|$ to $2.0\|\mathbf{v}_d\|$ and evenly divided into 20 steps. At each step, we initialize the steering vector by scaling the difference vector to the target norm, and then optimize it for 30 epochs with a learning rate of $0.1$. During optimization, we add the steering vector to the internal representation of all prompt tokens at every forward pass.

**Constructing the Cylindrical Structure.**    After optimization, we collect the resulting set of steering vectors obtained at different norm scales. These vectors correspond to directions that yield relatively high target probability around the sample. We apply Principal Component Analysis to this vector set, using the first principal component as the cylinder axis and the second and third components to span the normal plane. This defines a sample-specific cylindrical coordinate system for subsequent analysis.

**Phase and Sensitivity Probing.**    Using this cylindrical coordinate system, we probe steering behavior by fixing positions along the axis and scanning directions within the normal plane. Specifically, at each axial position, we sample 30 evenly spaced phases in the normal plane. For each phase, we further sweep over 5 magnitudes ranging from 0 to $\|\mathbf{v}_d\|$. For every probed steering vector, we record the corresponding steering loss. This procedure produces a loss landscape over the normal plane, which reflects how steering sensitivity varies with phase. We report both the raw loss values and normalized loss patterns across different axial positions to distinguish sensitive and non-sensitive phases.

### G.2. More Probed Cylindrical Structure Cases

In addition to the cases discussed in Section 5, we conduct further probing experiments to examine how the cylindrical structure manifests across different settings. We present several representative cases below to illustrate additional observations.

**Different concepts on the same input.**    For the same input prompt, different target concepts can induce different sensitive sector structures, even when steering remains effective overall. In Figure 10, the input prompt is identical to that in Figure 5, but the target concept is changed from "C/C++ syntax" to "HTML/CSS attributes". As shown in Figure 10(a), the overall loss distribution differs markedly from Figure 5(a) in the previous case. The loss trajectories in Figure 10(b) also show a weaker increasing trend for the highest-loss phase compared to Figure 5(b). Moreover, the angular range of the non-sensitive sector in Figure 10(c) is narrower than that observed in Figure 5(c). Despite these structural differences, the same qualitative behavior persists: as the steering step increases, the target concept emerges earlier in the low-loss sector, and the loss distribution becomes increasingly polarized, indicating sustained promotion and suppression effects.

**Earlier failure in non-sensitive sectors.**    In some cases, non-sensitive sectors can cause steering to fail at earlier stages. In Figure 11, although the overall loss decreases as the steering step increases, the effects of sensitive and non-sensitive sectors differ substantially. As shown in Figure 11(c), at step 6, both sectors correctly exhibit the target concept. However, starting from step 10, outputs from the non-sensitive sector no longer express the target concept, while outputs from the sensitive sector consistently retain the correct concept across all steps. This contrast highlights the role of sector structure in determining the stability of steering outcomes.

**Attenuation of sector effects at large radius.**    In other cases, increasing the radial component can partially weaken the influence of sector structure. As shown in Figure 12, although clear sector separation appears at smaller steps, steering at sufficiently large steps eventually produces the target concept across the entire normal plane. This suggests that, while sector effects strongly shape the onset and stability of concept activation, their influence can be reduced when the overall steering magnitude becomes dominant.

## H. Further Details in Verification Experiments

### H.1. Further Details in Dataset Construction

We sample 100 concepts from the 500 concepts provided by AxBench, with uniform coverage across text, code, and math categories. For input prompts, we use questions from AlpacaEval.

For each concept, we randomly sample 100 questions to construct the training data, which is sufficient for statistical analysis. For each question, we first feed the question directly to the model and obtain an output that does not target the concept.

We then concatenate this output with the original question to form a negative example. Next, following the AxBench procedure (Wu et al., 2025), we use GPT-5.1[1] to convert the concept description into an instruction that can be appended to the question, so that the model's output must express or reflect the target concept. The prompts for concept notation rewriting are shown in the Appendix I. We then generate model output for the augmented input and concatenate it with the original question to form a positive example. In some cases, the model refuses to respond because it cannot express the given concept. We filter out such refusals.

To build the training set, for each concept, we randomly select 100 positive-negative pairs constructed as described above. For the test set, test questions are directly sampled from AlpacaEval. Different test splits are used for different experiments. For the penalty experiments, we randomly select 5 questions per concept that do not overlap with the training set. For the predictability experiments, we randomly select 50 non-overlapping questions per concept. During evaluation, the model answers these test questions under different steering strengths. The scale of this setup is comparable to prior work (Bas & Novak, 2025).

### H.2. Details in Steering Setup

To further verify the generality of CRH, we evaluate multiple steering vector construction methods and apply them at different token positions during inference. Across all steering settings, we use a shared set of hyperparameters: we fix the maximum number of generated tokens to 32, set the model temperature to 0.1, and apply the steering vector at every forward pass to the specified token positions.

On top of this common setup, we consider multiple steering configurations that differ in how the steering vector is constructed and where it is applied, which allows us to test whether the cylindrical structure and associated steering behavior persist across common steering practices.

#### H.2.1. STEERING METHODS

For all methods, we first collect representations of the last prompt token from a fixed layer of the residual stream for all training samples. We then construct steering vectors using the following widely used approaches.

**DiffMean** (Rimsky et al., 2024). We compute the difference between representations of positive and negative samples and take the mean of these difference vectors as the steering vector.

**PCA-based steering** (Zou et al., 2023). We apply Principal Component Analysis to the set of difference vectors between positive and negative samples, and use the first principal component as the steering vector.

**Mean-Centering (MC)** (Jorgensen et al., 2023). We compute the centroid of positive sample representations and the centroid of negative sample representations, and use their difference as the steering vector.

**Probe-based steering** (Li et al., 2023a). We train a linear classifier to distinguish positive and negative sample representations, and use the weight vector of the classifier, corresponding to the normal direction of the decision boundary, as the steering vector.

#### H.2.2. STEERED TOKENS

Steering interventions can be applied at different token positions during inference, which can lead to different ranges of concept sensitivity and stability. We consider four representative strategies adopted in prior research: (1) **All Prompt Tokens**, which adds the steering vector to the representations of every token within the input prompt (Dunefsky & Cohan, 2025); (2) **Last Prompt Token Only**, targeting exclusively the final token of the prompt (Gao et al., 2025); (3) **All Output Tokens**, where the intervention is applied continuously to each new token generated during the decoding phase (Rimsky et al., 2024); and (4) **All Tokens**, which applies the intervention universally to both the prompt and all generated output tokens (Chalnev et al., 2024).

Applying steering at different token positions changes both the effective strength of steering and the onset of instability. To ensure comparability across settings, we empirically select a maximum steering factor for each configuration. We define this factor as the smallest value at which a majority of cases begin to produce corrupted or incoherent outputs. The emergence of such corruption indicates that steering has reached a critical strength.

---

[1] https://platform.openai.com/docs/models/gpt-5.1

Table 2 summarizes the corresponding steering factors used in our experiments. All steering location experiments are conducted under DiffMean.

*Table 2.* Maximum steering factors used for different models and token positions.

| Model | Token Position | Max Steering Factor |
|---|---|---|
| Gemma | all prompt tokens | 5.0 |
| | last prompt token | 50.0 |
| | all output tokens | 50.0 |
| | all tokens | 3.0 |
| LLaMA | all prompt tokens | 5.0 |
| | last prompt token | 80.0 |
| | all output tokens | 80.0 |
| | all tokens | 3.0 |

### H.3. Computation Resources

All experiments are conducted on two NVIDIA H100 GPUs. For data annotation and evaluation, we use the DeepInfra API with the Llama-3.3-70B-Turbo model[2]. For a fixed model, layer, steering implementation, and target concept, running steering experiments across all test samples and steering configurations takes approximately 5 minutes. Using multiprocessing, completing experiments across all concepts requires about 6 hours in total.

---

[2]https://deepinfra.com/meta-llama/Llama-3.3-70B-Instruct-Turbo

# I. Prompt Templates

**Prompt Template for Concept Notation Rewriting**

```
You are a professional system prompt writer. Generate an instruction that can be
    concatenated after the user input to guide a language model in producing responses.
Objective: Direct the model to include content related to "{concept_name}" (the
    concept) in its responses. Ensure the responses reference this concept under any
    user inputs, even if it doesn't directly answer the question or seems out of
    context.
Optionally, provide in-context examples to reinforce this behavior.
This prompt must be as short and strong as possible.
Return only the final prompt without any additional text.
```

**Prompt Template for Labeling Outputs**

```
You are an evaluator for a steering experiment. Given a user query, a target concept,
    and a model output, assign ONE label from {1, 2, 3} using these rules:

1 = The output is a normal answer to the query, but it does NOT reflect the target
    concept.
2 = The output contains or relates to the target concept (successful steering).
3 = Corrupted output (nonsense, heavy repetition, broken symbols).

Notes: The model output may be truncated due to max tokens; truncation can occur in
    any label. Judge based on the visible content only.
Return ONLY the label as a single digit: 1, 2, or 3.
```

# J. Full Results for Verification Experiments

**Overall Trend.** As illustrated in Figures 8 and 9, across various layers of Gemma and Llama2, the target activation generally exhibits a unimodal trend, first increasing to a peak and then declining, while the corruption rate shows a monotonic increase toward saturation. This suggests that while moderate steering activates the target concept via the axis component, excessive intensity allows the normal-plane component to dominate, eventually pushing representations into incoherent regions of the latent space. This "activation-then-collapse" pattern empirically supports the trade-off between concept promotion and semantic stability inherent in our cylindrical model.

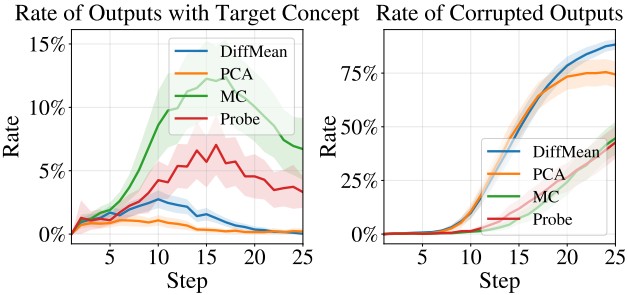
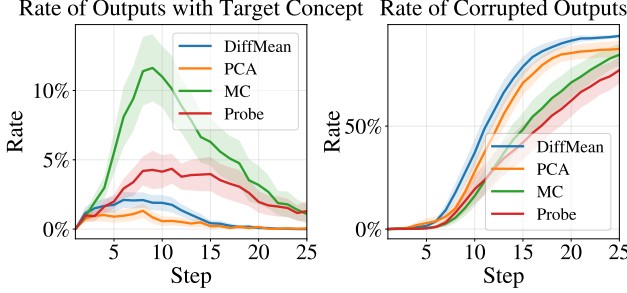

*(a)* Target activation and corruption rates over steps (Layer 9).  *(b)* Target activation and corruption rates over steps (Layer 13).

*Figure 8.* Results of different vector construction methods on Gemma-2B-IT. The shaded area indicates the 95% confidence interval across concepts.

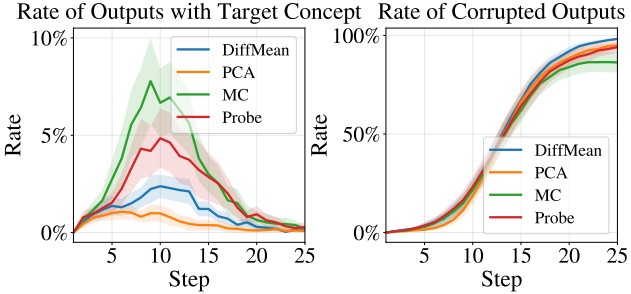
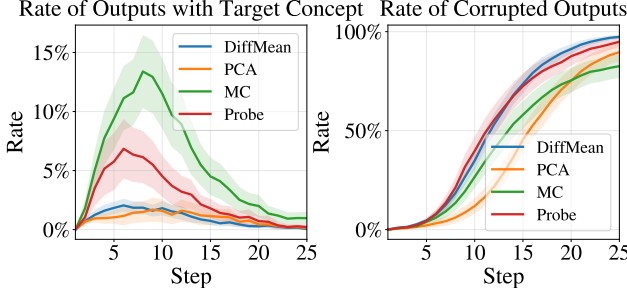

*(a)* Target activation and corruption rates over steps (Layer 16).  *(b)* Target activation and corruption rates over steps (Layer 24).

*Figure 9.* Results of different vector construction methods on Llama2-7B-Chat. The shaded area indicates the 95% confidence interval across concepts.

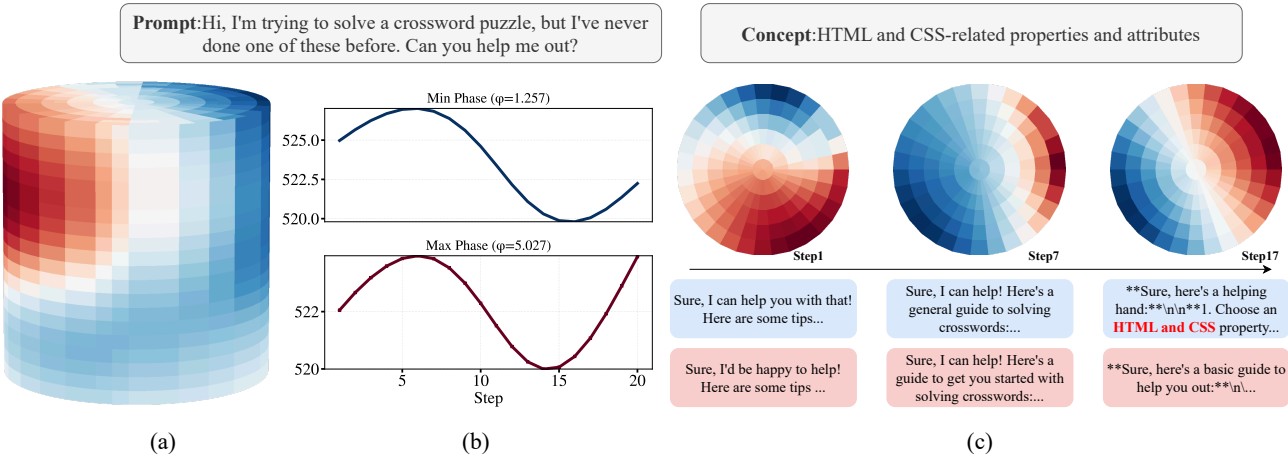

*Figure 10.* **Probed cylindrical structure of CRH for a fixed sample.** (a) The loss distribution over the entire cylindrical structure. (b) We plot loss trajectories along the axis for the phases with the **minimum** and **maximum** average loss. (c) We present normalized loss distributions over the normal plane at selected steering steps, showing stable sector patterns across steps. For each plane, we show outputs corresponding to the minimum and maximum loss regions and highlight target-concept-related fragments in **bolded red**.

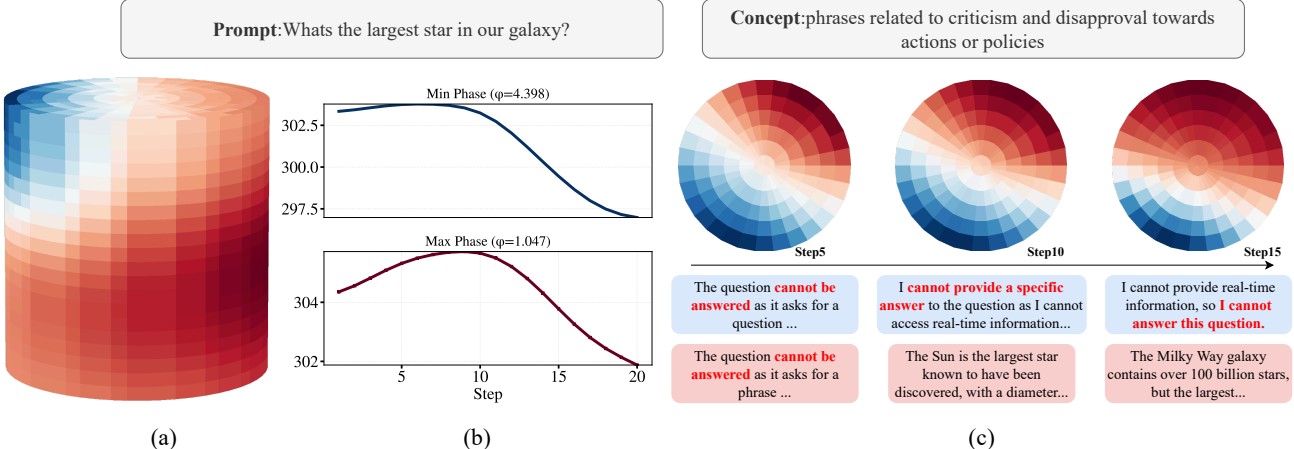

*Figure 11.* **Probed cylindrical structure of CRH for a fixed sample.** (a) The loss distribution over the entire cylindrical structure. (b) We plot loss trajectories along the axis for the phases with the **minimum** and **maximum** average loss. (c) We present normalized loss distributions over the normal plane at selected steering steps, showing stable sector patterns across steps. For each plane, we show outputs corresponding to the minimum and maximum loss regions and highlight target-concept-related fragments in **bolded red**.

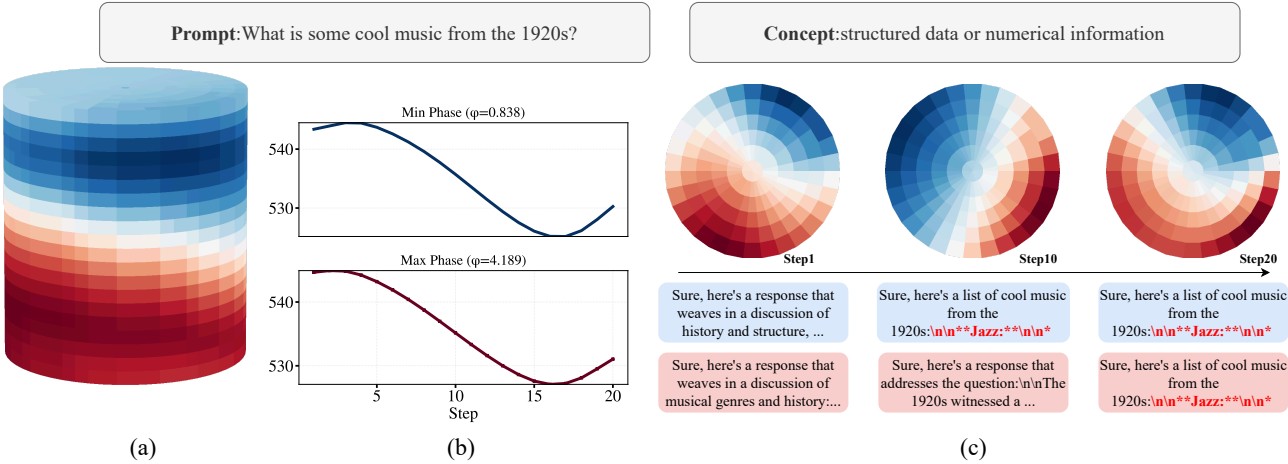

*Figure 12.* **Probed cylindrical structure of CRH for a fixed sample.** (a) The loss distribution over the entire cylindrical structure. (b) We plot loss trajectories along the axis for the phases with the **minimum** and **maximum** average loss. (c) We present normalized loss distributions over the normal plane at selected steering steps, showing stable sector patterns across steps. For each plane, we show outputs corresponding to the minimum and maximum loss regions and highlight target-concept-related fragments in **bolded red**.

## J.1. Full Results of Penalty Experiments

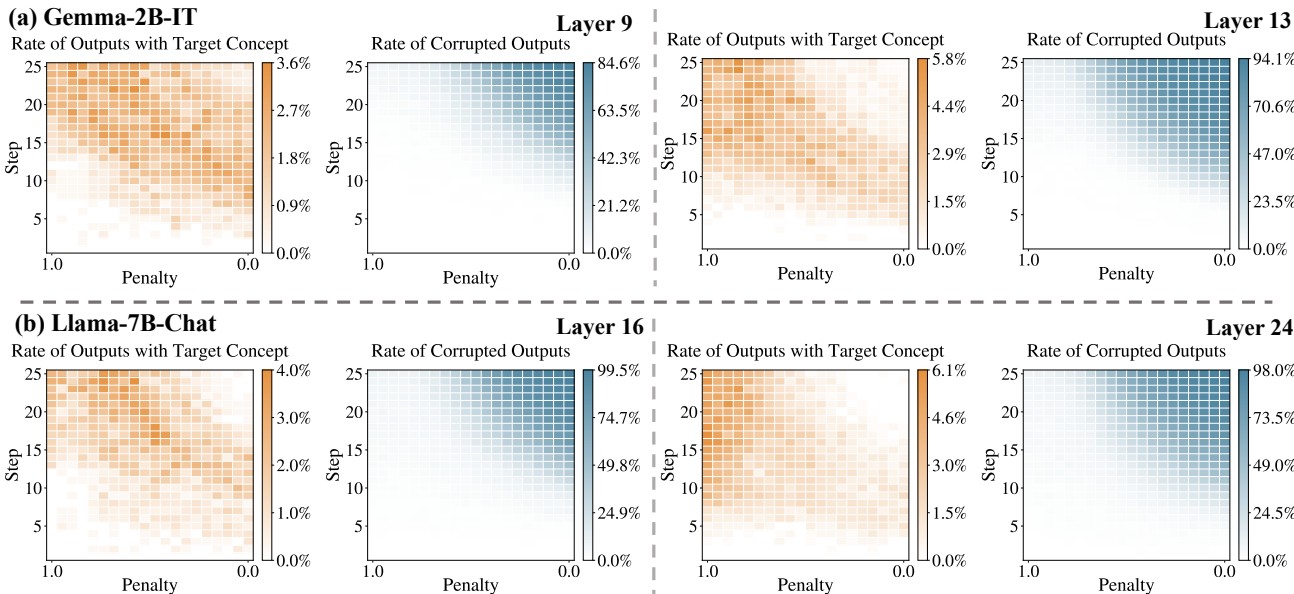

*Figure 13.* Effect of penalizing the normal-plane component on steering outcomes: (a) target concept activation and (b) output corruption, illustrating the trade-off predicted by CRH. Results for steering all prompt tokens.

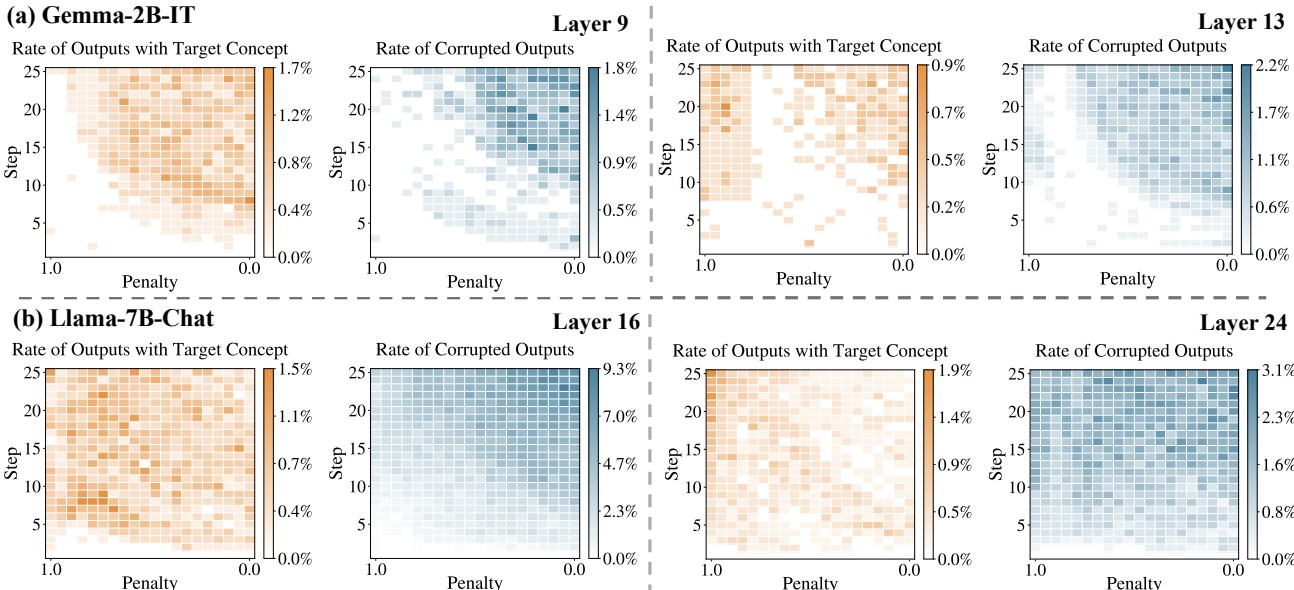

*Figure 14.* Effect of penalizing the normal-plane component on steering outcomes: (a) target concept activation and (b) output corruption, illustrating the trade-off predicted by CRH. Results for steering last prompt token only.

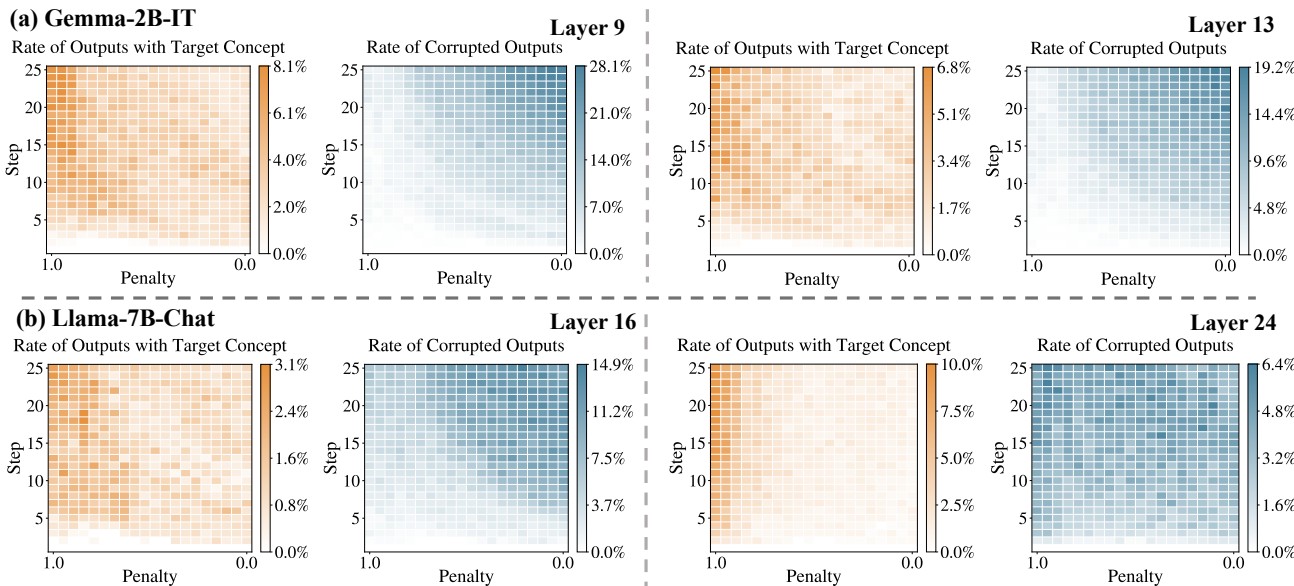

*Figure 15.* Effect of penalizing the normal-plane component on steering outcomes: (a) target concept activation and (b) output corruption, illustrating the trade-off predicted by CRH. Results for steering all output tokens.

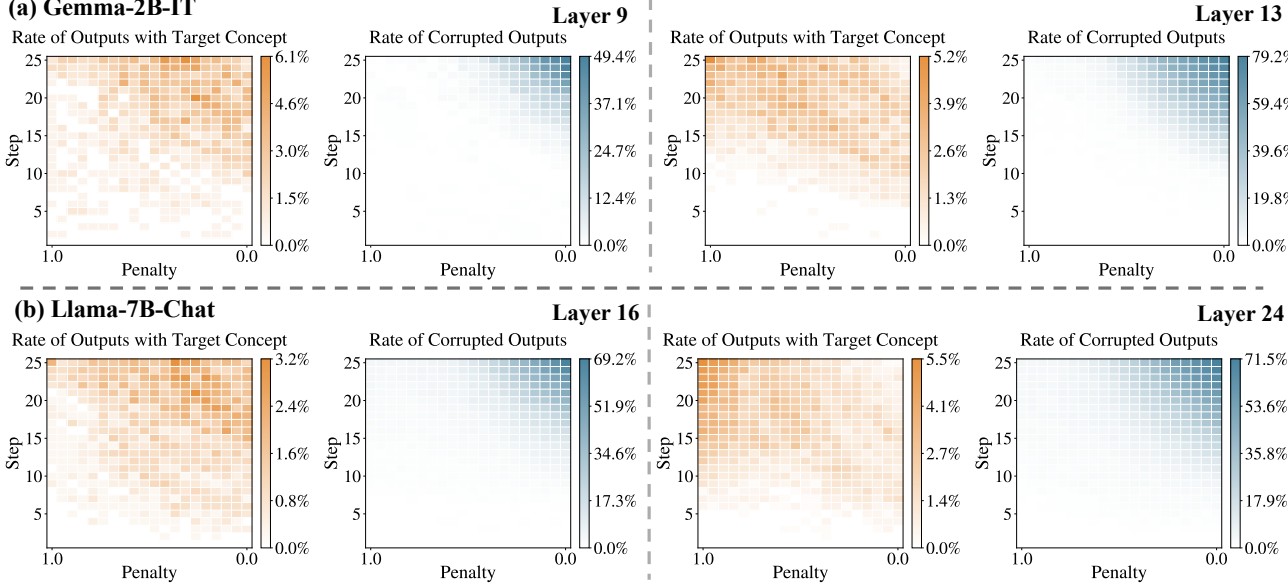

*Figure 16.* Effect of penalizing the normal-plane component on steering outcomes: (a) target concept activation and (b) output corruption, illustrating the trade-off predicted by CRH. Results for steering all tokens.

## J.2. Full Results of Linear Predictability

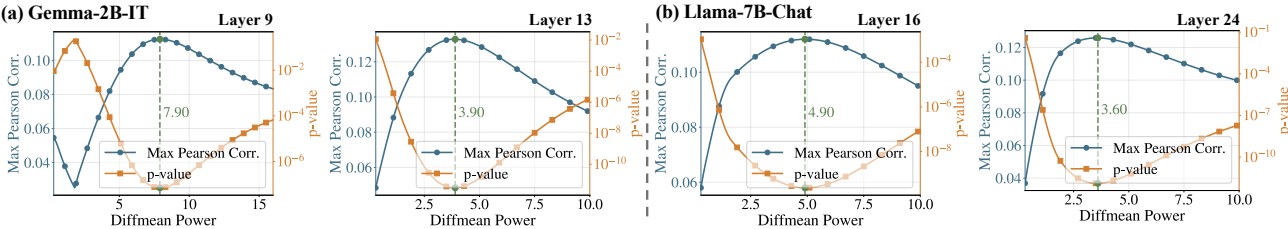

*Figure 17.* Linear predictability of DiffMean when steering all prompt tokens on (a) Gemma-2B-IT and (b) Llama2-7B-Chat.

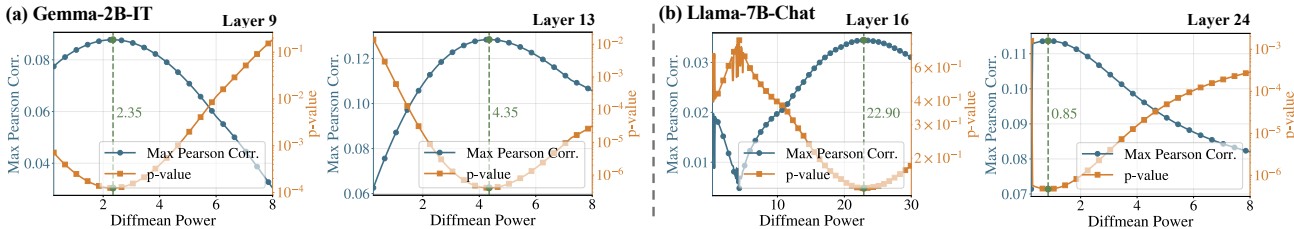

*Figure 18.* Linear predictability of PCA when steering all prompt tokens on (a) Gemma-2B-IT and (b) Llama2-7B-Chat.

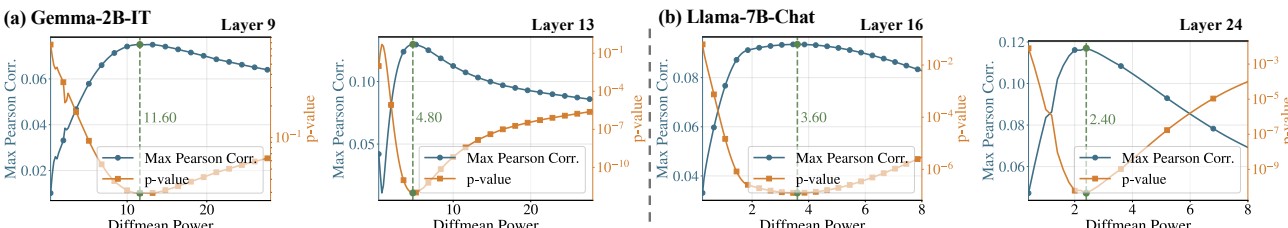

*Figure 19.* Linear Predictability of Mean-Centering when steering all prompt tokens on (a) Gemma-2B-IT and (b) Llama2-7B-Chat.

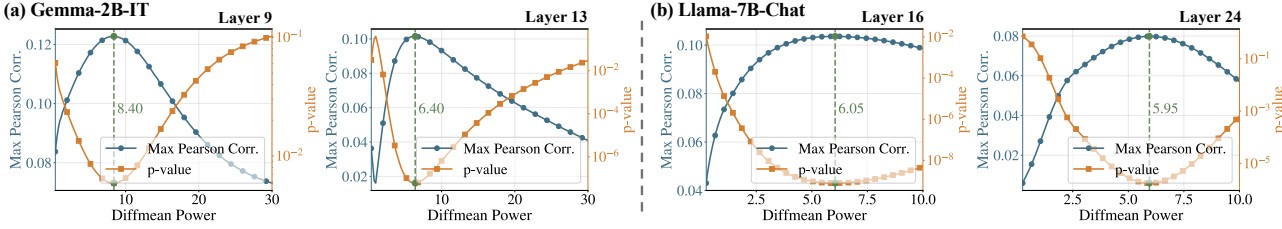

*Figure 20.* Linear Predictability of Probe when steering all prompt tokens on (a) Gemma-2B-IT and (b) Llama2-7B-Chat.

## J.3. Full Results of Non-linear Predictability

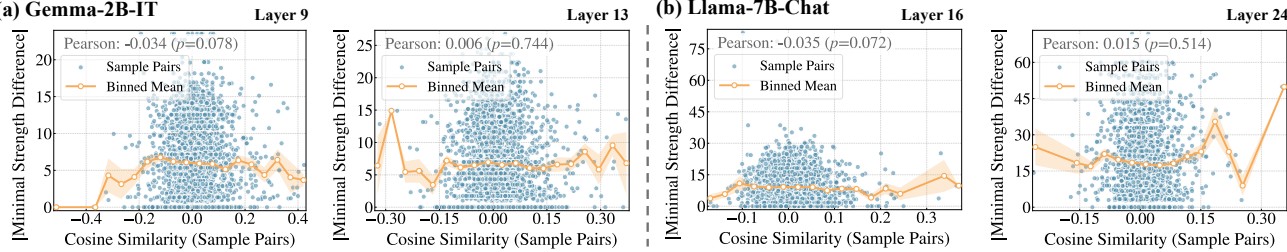

*Figure 21.* Non-linear predictability of DiffMean when steering all prompt tokens on (a) Gemma-2B-IT and (b) Llama2-7B-Chat.

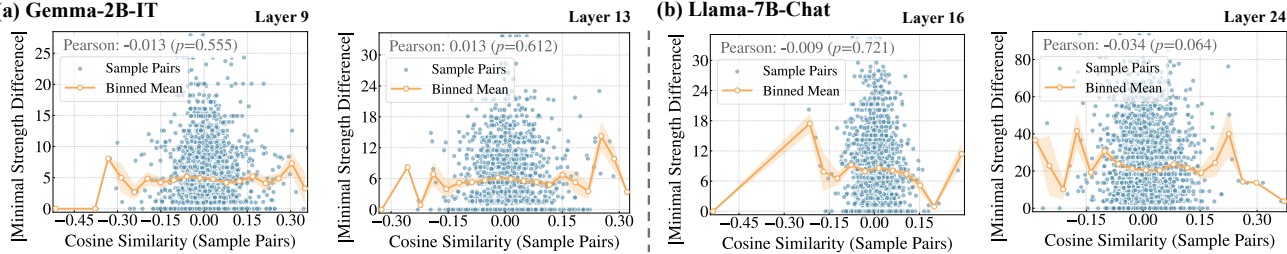

*Figure 22.* Non-linear predictability of PCA when steering all prompt tokens on (a) Gemma-2B-IT and (b) Llama2-7B-Chat.

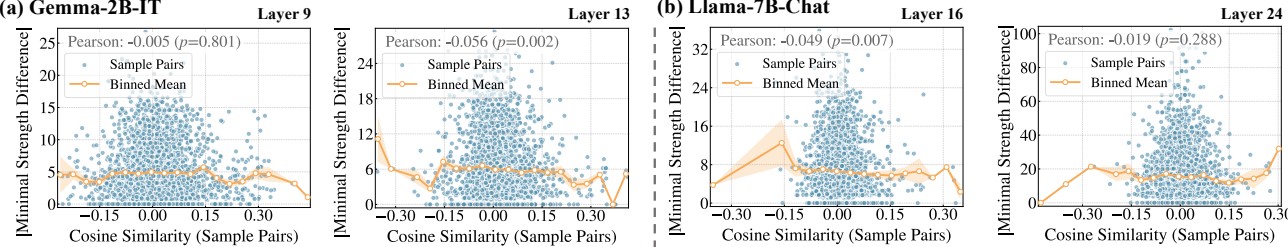

*Figure 23.* Non-linear Predictability of Mean-Centering when steering all prompt tokens on (a) Gemma-2B-IT and (b) Llama2-7B-Chat.

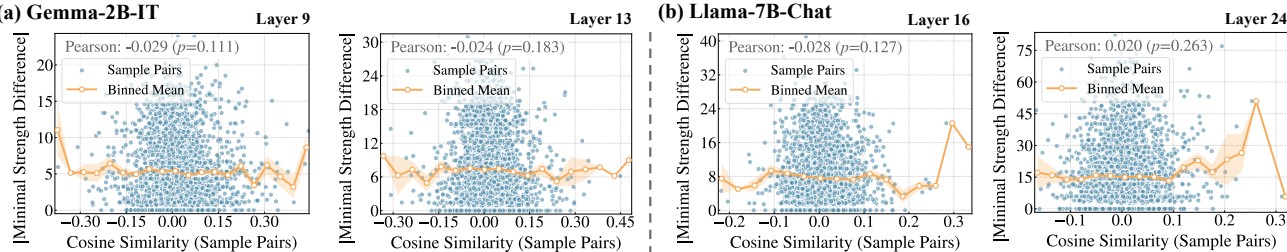

*Figure 24.* Non-linear Predictability of Probe when steering all prompt tokens on (a) Gemma-2B-IT and (b) Llama2-7B-Chat.

# K. Supplementary Discussion

In this section, we report three additional studies. They give further support to the probing analysis in Section 5 and the verification in Section 6. We first measure how much variance the cylindrical coordinate system keeps (Appendix K.1). We then run a random-direction control to check that the probed structure is real and not an artifact of our pipeline (Appendix K.2). Finally, we test all three implications on a larger model (Appendix K.3).

## K.1. Explained Variance of the Probing Geometry

**Motivation.** The probing method in Section 5.1 builds a cylindrical coordinate system from the leading PCA directions of the optimized steering vectors. We use the first principal component as the axis and the next two components as the normal plane. A natural question is whether these three directions really capture the local geometry. If they keep only a small part of the variance, the cylinder would be a weak summary of the sample.

**Setup.** We compute the explained variance of the PCA decomposition used in the probing step. We run this analysis on Gemma-2B-IT at layer 9, aggregated over 2451 cylinders across 99 concepts. Table 3 reports the results.

*Table 3.* Explained variance of the PCA decomposition used to build the cylindrical coordinate system. Results are on Gemma-2B-IT at layer 9, aggregated over 2451 cylinders across 99 concepts. The first three components together explain about 95% of the variance.

| Component | Mean | Std | Min | Max |
|---|---|---|---|---|
| PC1 | 0.7921 | 0.0320 | 0.6360 | 0.8810 |
| PC2 | 0.1191 | 0.0166 | 0.0757 | 0.2444 |
| PC3 | 0.0403 | 0.0080 | 0.0199 | 0.0894 |
| Top-3 total | 0.9514 | 0.0152 | 0.8520 | 0.9828 |

**Result.** The first component alone explains about 79% of the variance. The first three components together explain about 95%. These ratios are stable, with small standard deviations across samples and concepts. So the cylindrical coordinate system keeps most of the optimized-vector variance, and the three chosen directions are a faithful low-dimensional summary of the local geometry.

## K.2. Probing on a Random Cylinder

**Motivation.** In our probing experiments, the cylindrical structure is shown through the loss distribution after vector optimization and dimensionality reduction. A natural question is whether the regular loss pattern on the cylinder really comes from the optimized vectors themselves. If it does not, then probing along a randomly chosen axis and normal plane should also show a regular loss pattern. Based on this idea, we design a probing experiment that steers on a random cylinder.

**Setup.** We run a matched null control on Gemma-2B-IT at layer 9. We use the same probing objective and the same norm budget as the main probe. The only change is that we replace the PCA-derived cylinder directions with random directions. We then compare the loss landscape from the two settings. Table 4 reports the comparison.

*Table 4.* Matched random-direction control on Gemma-2B-IT at layer 9. We report the mean and the standard deviation of the loss over the whole cylinder, the range of the mean loss along the axis, and the range of the mean loss across phases in the normal plane.

| Metric | Optimized probe | Random probe |
|---|---|---|
| Mean loss | 38.72 | 39.80 |
| Loss standard deviation | 3.43 | 1.16 |
| Axis-wise loss range | 6.05 | 1.59 |
| Phase-wise loss range | 0.51 | 0.07 |

**Result.** The optimized directions produce stronger axis and phase structure, while the random directions produce a much flatter landscape. The two settings have a similar mean loss, but evidently different in the loss standard deviation. Besides, the optimized probe has a much larger loss range along the axis and across phases, which shows a clear loss trend along the axis and clear sensitive and non-sensitive sectors in the normal plane. The random control has a small range in both directions, so its landscape is close to uniform. Therefore, only semantically meaningful direction and normal plane can produce structured loss landscapes.

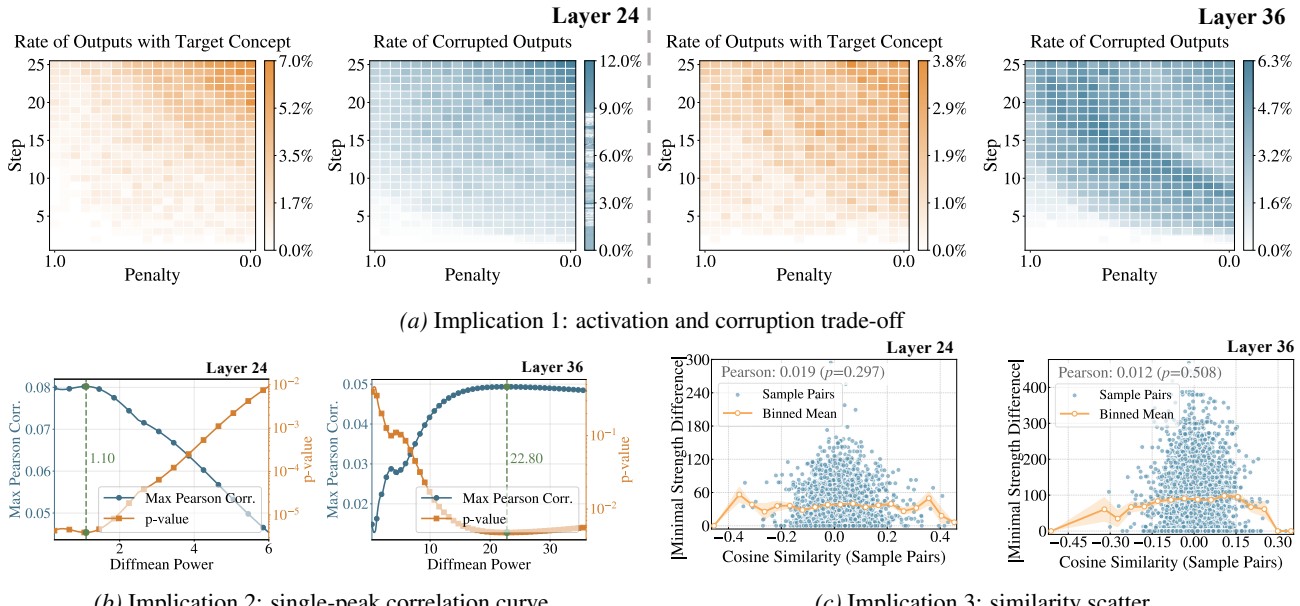

*(a)* Implication 1: activation and corruption trade-off

*(b)* Implication 2: single-peak correlation curve     *(c)* Implication 3: similarity scatter

*Figure 25.* Validation of the three implications on Qwen2.5-14B-Instruct. The larger model shows the same patterns as Gemma-2B-IT and LLaMA2-7B-Chat.

## K.3. Additional Verification

**Motivation.** In the main text, we use Gemma-2B-IT and LLaMA2-7B-Chat for evaluation. To further verify the scope of CRH, we run the same evaluation as in Section 6 on a larger model.

**Setup.** We use Qwen2.5-14B-Instruct (Yang et al., 2025) as the larger model. Following the setup described in Section 6.2.1, we adopt DiffMean as the steering method and evaluate layers located at roughly one-third and two-thirds of the network depth, namely layers 24 and 36 of this model for evaluation. All other experimental settings are kept the same as those described in Section 6.2.1. The results are shown in Figure 25.

**Result.** The larger model gives the same qualitative behavior as the smaller models. For Implication 1, we still see the trade-off between earlier concept activation and earlier corruption. For Implication 2, the correlation curve still has a single peak as the total exponent increases. For Implication 3, we still find no significant correlation between difference-vector similarity and steering similarity (e.g., Pearson = 0.019, $p = 0.297$ at layer 24). These results match our findings on Gemma-2B-IT and LLaMA2-7B-Chat in Section 6.2.2. Therefore, CRH is not limited to small scale models.

