# OpenReview forum: "The Cylindrical Representation Hypothesis for Language Model Steering"
_ICML.cc/2026/Conference — ICML 2026 regular_

### Official Review · Reviewer_2GP1 · 2026-03-11

**Soundness:** 3
**Presentation:** 3
**Significance:** 2
**Originality:** 3
**Overall Recommendation:** 4
**Confidence:** 2

**Summary:**

The paper introduces the cylindrical representation hypothesis (CRH), which speculates that steering failures in LLMs are due to the linear representation hypothesis not correctly accounting for non-orthogonality between features. The CRH hypothesizes that steering may fail to work on a sample-by-sample basis due to different effects from overlapping features causing the "correct" steering direction to change per-token, breaking the orthogonal space into "steerable" and "non-steerable" sectors. These sectors cannot be found during steering though, limiting the applicability of the hypothesis. The paper then validates that CRH-esque effects are seen in steering experiments.

**Compliance With Llm Reviewing Policy:**

Affirmed.

**Final Justification:**

I find it hard to believe this is a serious problem for steering models, and am not convinced that error in extracting steering vectors is not the more likely culprit, but the other reviewers all gave this paper a 5 so I assume I must be missing something that's obvious to everyone else. I am thus raising my score to 4 but lowering my confidence to give the authors the benefit of the doubt.

**Key Questions For Authors:**

- Does knowledge of the CRH lead to any different method or technique to improve steering or probing or a concrete task?
- How do you know that the reason current steering doesn't work well isn't just that we're finding noisy steering vectors? E.g. if a current steering vector is only 50% correct and 50% random junk, wouldn't that also explain a lot of the incorrect steering outcomes we're seeing?
- How much superposition needs to exist before CRH steering effects begin to matter? Wouldn't the model be incentivized to minimize this interference for commonly co-occurring features anyway?

**Limitations:**

yes

**Strengths And Weaknesses:**

## Strengths

- The CRH makes sense mathematically, that overlapping features will cause some interference.
- The description of the CRH is well-written.
- The experiments are consistent with the CRH being true (although I'm not completely convinced it's the only explanation)

## Weaknesses

- The paper doesn't address how much superposition is necessary before these CRH effects become significant. If the level of non-orthogonality is small, maybe this isn't a big deal?
- LLMs are likely incentivized to make commonly co-occurring features more orthogonal, since this overlap is also a problem for the LLM making use of features itself. I think that toy models of superposition work shows the model strategically reducing superposition where it is most likely to occur. The paper doesn't seem to engage with this idea.
- The paper claims that the LRH requires all features to be perfectly orthogonal, but this is not true. Even the old "toy models of superposition" work introducing the LRH explicitly talks about features being not completely orthogonal.
- Since the steerable and non-steerable sectors cannot be known at inference time, it seems like there's not a different or better method of steering that's proposed, so the usefulness of this idea seems low.
- It seems like a lot of the failure modes predicted by the CRH could also be explained by the steering vectors we're finding just not being entirely correct (e.g. maybe 50% correct direction, 50% random junk).

---

> ### Author Rebuttal · Authors · 2026-03-29
>
> Thank you for the detailed review. We appreciate that you found the core CRH idea mathematically sensible, that you found the description clear, and that you agreed the experiments are at least consistent with CRH.
>
> **w1:how much superposition is needed before CRH matters & c3:threshold of interference.** Thank you for raising this point. We think there is a slight misunderstanding about our goal.
>  The current paper does not try to estimate a universal threshold of “how much overlap is enough.” Doing that well would require a largely different setup, which is beyond the scope of the current paper. Besides, We have detailedly discussed  relationship between CRH and LRH, which is closely connected to the superposition theory.
>
> What we can say empirically is that the cases where CRH matters is not rare in current steering practice. Empirically, our extensive experiments show  that, across many concepts, models, layers, and steering settings, we repeatedly observe strong sample-specific irregularity and the trade-off between earlier concept activation and earlier corruption, which are consistent with the predictions of CRH. This suggests that the effects described by CRH are relevant in practical steering settings. However, we agree that estimating a precise threshold of interference is an interesting direction for future exploration. We will clarify this in the revision.
>
> **w2:models may reduce superposition for common features & w3:our claim about LRH and orthogonality.** Thank you for this point. We think there is a small misunderstanding.
> We do not assume that features in the activation space needs to be perfectly orthogonal, and LRH itself also does not require strict orthogonality in the raw space. In fact, in Section 2.1 we indicate that, LRH introduces a causal inner product, under which logically independent concepts can be treated as orthogonal for pure intervention. This can be viewed as orthoganality in a transformed space, not the original space.
>
> **w4:if sensitive sectors are not observable, is CRH useful? & c1:does CRH lead to a concrete method or task?** Thank you for pushing on usefulness. We see CRH as useful in at least two concrete ways already.
>
> First, CRH changes the diagnostic target. Under a pure “correct direction plus noise” view, one would expect that better denoising or better average alignment should largely remove the problem. CRH says the harder part is sample-specific phase interaction in a local geometry. This helps explain why steering can remain irregular even when the direction looks reasonable in aggregate.
>
> Second, CRH encourages new evaluation and steering methods. The penalty experiment shows that the axis-aligned component and the orthogonal component play different roles: one promotes concept emergence, while the other also drives instability. This means future steering methods should not optimize only for average concept activation. They should explicitly reason about the gain-risk trade-off and about sample-specific steerability. In the paper, we point to steerability prediction as the next step. We agree that the current submission is mainly a framework paper, not a finished new steering algorithm, and we will state that more plainly. But we do believe CRH gives a concrete direction for better probing, better diagnostics, and eventually better steering methods.
>
> **w5: could the failures just come from noisy steering vectors? & c2: is it just random junk in the vector?** Thank you for raising this important alternative explanation.
>
> We believe the issue is not well explained by a simple “correct direction + random noise” view. The main limitation of this view is that it cannot explain the strong sample-specific behavior observed in practice. We have discussed in details in Section 2.2.
>
> Furthermore, in our paper, Implication 3 is an evidence against this view. If the noise hypothesis were correct, then vectors with similar directions should contain similar noise components, and therefore lead to similar steering behavior (line 373). However, this is not what we observe. In Figure 7(b), the cosine similarity between difference vectors does not correlate with similarity in steering outcomes (Pearson ≈ 0). This result directly contradicts the noise-based explanation.
>
> In addition, prior work has shown that LRH-based methods for predicting steerability can fail when the experimental setup changes (para.2, introduction). This suggests that the “clean direction + noise” view is not sufficient to explain steering behavior.
> Our goal is to provide a more structured explanation for these phenomena. CRH attributes and proves the instability to sample-specific geometry, rather than treating it as unstructured noise.
>
> We will make this comparison to the noise-based explanation clearer in the revision.
>
> ---
> *If our responses resolve your concerns, we would be very grateful if you could consider raising your score.*

---

> > ### Author Rebuttal · Reviewer_2GP1 · 2026-04-01
> >
> > Thank you for the responses. I'm still a bit confused about a few points:
> >
> > - Why wouldn't the model place concepts in a way that this interference is minimized? If I understand correctly, this same interference would be a big problem for the model as well, since it would need to calculate a different direction for each concept based on which other concepts are active in the given token, and this seems like something the model would avoid if possible. Or is this not a problem for the model as well, and only affects our attempts to steer the model?
> > - I understand you're not estimating how much overlap is enough that these effects become significant, but do you have any thoughts on this? Is it correct that the less overlap there is the less of a problem this becomes? My understanding is that in a 8000+ dim space commonly used in LLMs almost any reasonable number of concepts (e.g. trillions) can be fit with each direction having well under 0.1 cosine similarity max with all other concepts. And that's not even taking into account that the model can place concepts so as to minimize interference where necessary. This seems like such a small cosine similarity between concepts that I find it hard to believe that non-orthogonality is the core problem facing steering, rather than our techniques simply finding an off-target direction to steer to begin with, or the problem being related to manifolds or something.

---

> > > ### Author Response · Authors · 2026-04-08
> > >
> > > Thank you for these thoughtful questions. We appreciate the intuition behind them.
> > >
> > > **Regarding the first point (why the model does not avoid this interference):**
> > > We agree that models have an incentive to reduce harmful interference, and high-dimensional spaces allow many directions to be nearly orthogonal. This helps the model maintain stable behavior during inference.
> > >
> > > However, steering operates differently from normal inference. It applies a single external direction to the model to change its outputs. In CRH, each sample contains a different combination of concepts, so the same direction acturally interacts with different local structures. As described in our framework, this leads to different decompositions and thus different outcomes. From this perspective, we believe that the effect is not a major issue for the model’s own computation, but becomes visible when an external direction is applied across diverse inputs.
> > >
> > > **Regarding the second point (whether less overlap reduces the problem):**
> > > Thank you and we understand this intuition. Regarding the failure of steering, if the direction is extracted properly, less overlap indeed can improve the effectiveness. In the paper, our focus is broder. We investigate the *sample-specific nature of steering*: the success and failure of steering is not consistent and predictable currently. We have two clues in the paper to support this:
> > >
> > > 1. In the LRH view, higher separability is often associated with better steerability. However, prior work (as discussed in our introduction)  shows that separability does not reliably predict steering performance.
> > >
> > > 2. We also test the alternative explanation that unpredictability comes from noise. In Implication3: if this were the case, similar directions should lead to similar steering behavior, as they should contain comparable amount of noise. Empirically, we do not observe such a correlation. This suggests that the issue cannot be fully explained by overlap or noise.
> > >
> > > So, even with small pairwise cosine similarity, each sample is still a combination of multiple concepts, and these combinations vary. This leads to different local structures and thus different steering outcomes.
> > >
> > > Overall, reducing global overlap may influence representations, but it does not remove the sample-specific variability observed in practice. CRH is intended to explain this aspect.
> > >
> > > ---
> > >
> > > *Again, we thank the reviewer's dedication and effort for this paper.*

---

### Official Review · Reviewer_q9ib · 2026-03-13

**Soundness:** 3
**Presentation:** 3
**Significance:** 3
**Originality:** 3
**Overall Recommendation:** 5
**Confidence:** 3

**Summary:**

This paper primarily highlights the limitations of the Linear Representation Hypothesis and proposes the Cylindrical Representation Hypothesis, which posits that the concept difference vectors in LLM representations induce a cylindrical geometric structure around each sample, consisting of a central axis, a normal plane, and sensitive sectors. This provides a geometry-grounded explanation for the instability observed in activation steering.

**Compliance With Llm Reviewing Policy:**

Affirmed.

**Final Justification:**

I maintain a positive score after rebuttal.

**Key Questions For Authors:**

N/A

**Limitations:**

Yes

**Strengths And Weaknesses:**

## Strengths

1. The critique of LRH is well-founded.

2. Theorem 4.1 and Theorem 4.3 establish an interesting asymmetry. The magnitude of the normal-plane component is predictable, whereas the sensitive sector is not. This provides a theoretical basis for why steering behavior is observable in aggregate yet remains unpredictable at the individual sample level.

3. The experimental seeting is systematic, covering many concepts, two model architectures, multiple steering methods, and detailed analysis.

## Weaknesses

I am not an expert in the theoretical aspects, so I cannot identify any particularly glaring weaknesses on that front.

---

> ### Author Rebuttal · Authors · 2026-03-29
>
> Thank you for the positive evaluation. We especially appreciate your recognition about CRH.
>
> We used the rebuttal period to further strengthen the empirical evidence and clarity of the paper. The following additions directly improve the robustness and interpretability of the proposed CRH framework.
>
> Specifically, we conducted three targeted experiments to further support the cylindrical structure hypothesis and to address potential concerns about methodological artifacts and generality.
>
> ---
>
> **1. PCA Variance Explanation for Probing Geometry**
>
> We report the explained variance of the PCA decomposition used to construct the cylindrical coordinate system in Section 5.
>
> | Metric | Mean | Std | Min | Max |
> |------|------|------|------|------|
> | PC1 Variance Ratio | 0.7921 | 0.0320 | 0.6360 | 0.8810 |
> | PC2 Variance Ratio | 0.1191 | 0.0166 | 0.0757 | 0.2444 |
> | PC3 Variance Ratio | 0.0403 | 0.0080 | 0.0199 | 0.0894 |
> | **Top-3 Total** | **0.9514** | **0.0152** | **0.8520** | **0.9828** |
>
> - Setup: Gemma-2B-IT, layer 9, aggregated over 2451 cylinders across 99 concepts.
> - Observation: The first three principal components consistently explain over 95% of the variance.
> - Implication: This supports the use of a low-dimensional cylindrical structure as a summary of the local geometry.
>
> ---
>
> **2. Matched null control for the probing pipeline**
>
> We introduce a matched null control by replacing the PCA-derived directions with random directions under the same probing setup.
>
> | Metric | PCA-based | Random Control |
> |------|-----------|----------------|
> | Loss Std (phase variation) | 2.52 | 0.67 |
> | Axis Loss Range | 6.05 | 1.59 |
> | Angle Variation Range | 1.46 | 0.49 |
> | Statistical Significance | p < 1e-4 | not significant |
>
> Additional observations:
>
> - Axis structure:
>   - PCA: clear monotonic loss trend along axis
>   - Random: weak and unstable variation
>
> - Phase structure:
>   - PCA: strong variation across angles
>   - Random: nearly flat response
>
> - Conclusion:
>   Under identical settings, PCA-derived directions recover clear structure, while random directions do not.
>
> ---
>
> **3. Validation on a larger model (Qwen2.5-14B)**
>
> We further test all three implications on a larger model, Qwen2.5-14B.
>
> | Aspect | Observation |
> |------|-------------|
> | Implication 1 | Same trade-off between earlier activation and earlier corruption |
> | Implication 2 | Same unimodal trend |
> | Implication 3 | No significant correlation (Pearson = 0.019, p = 0.297) |
>
> - Conclusion: Similar patterns are observed on a larger model, consistent with our main findings.
> - We provide the full results in our anonymous GitHub repository (folder: `appl_results_qwen2.5_14b`).
>
> ---
>
> We will further improve presentation clarity in the revision:
> (1) Simplify Section 3.2.
> (2) Remove the duplicated paragraph in Section 5.1.
> (3) Add clearer descriptions of the probing setup.
>
> These additions strengthen the empirical support for CRH and improve clarity.
>
> ---
>
> *If these additional results strengthen your confidence in the paper, we would be grateful if you could consider increasing your confidence score.*

---

> > ### Author Rebuttal · Reviewer_q9ib · 2026-04-02
> >
> > Thank you to the authors for the additional empirical evaluations. I will increase my confidence accordingly. Good luck!

---

> > > ### Author Response · Authors · 2026-04-08
> > >
> > > Dear Reviewer q9ib,
> > >
> > > Thank you for your continuous support for the paper!
> > >
> > > Best,
> > >
> > > the Authors

---

### Official Review · Reviewer_gBDL · 2026-03-13

**Soundness:** 3
**Presentation:** 3
**Significance:** 3
**Originality:** 3
**Overall Recommendation:** 5
**Confidence:** 4

**Summary:**

The submission introduces the Cylindrical Representation Hypothesis (CRH) as a refinement of the Linear Representation Hypothesis for activation steering in language models. The starting point is simple but important: once the number of relevant concepts exceeds the dimensionality of the representation space, the orthogonality story behind LRH cannot hold exactly, so interference is unavoidable. CRH keeps the linearity assumption but drops the requirement that concepts be orthogonal. In its place, the paper argues for a sample-specific geometry consisting of a central axis, a normal plane, and sensitive versus non-sensitive sectors within that plane. The key theoretical claim is that the magnitude of the normal-plane component is predictable from observable structure, whereas the sector that determines success or failure is not. That gives the authors a concrete explanation for a familiar empirical pattern in steering: aggregate trends are often measurable, but individual outcomes remain brittle. The experiments test this picture on Gemma-2B-IT and LLaMA2-7B-Chat across 100 concepts, multiple layers, and several steering constructions.

**Compliance With Llm Reviewing Policy:**

Affirmed.

**Final Justification:**

The rebuttal addressed the issue. The paper should be accepted.

**Key Questions For Authors:**

- First, have you run any controls to test whether the sector structure in Section 5 survives outside the PCA-on-optimized-vectors setup? A matched random-vector or null-task control would make the probing evidence much more convincing.

- Second, how sensitive is Figure 7(a) to the choice of steerability metric? Since the appendix suggests a nonlinear, often unimodal activation curve, I would like to see the same analysis with at least one alternative metric.

- Third, can you give either a stronger formal argument or a cleaner empirical one for why the idealized output-space theory should transfer to intermediate residual-stream states? For example, does the strength of the predicted correlation systematically improve as one moves closer to the output layers?

- Fourth, have you checked whether the same patterns appear in a larger model? Since the entire motivation leans on interference under limited representational degrees of freedom, scale seems especially relevant here.

**Limitations:**

- The appendix does include a reasonable limitations section, and I appreciated that the authors explicitly present CRH as a conceptual framework rather than a directly observed property. I would still encourage them to foreground one additional limitation more clearly in the main paper: the mismatch between the idealized space used in the theory and the hidden-state space used in the experiments is not a side note; it is one of the central interpretive bottlenecks in the paper.

- My takeaway is that there is a real idea here, and in places the paper is genuinely insightful. The main thing holding it back for me is that the strongest empirical evidence for the cylindrical picture is also the most vulnerable to alternative explanations. Tightening that part, and being more precise about the theory-to-experiment bridge, would make the paper much more convincing.

**Strengths And Weaknesses:**

Strengths

- The motivation in Section 2 works well. The dimensionality argument against lossless orthogonal control is not complicated, but it is exactly the right place to start, and the three-concepts-in-2D counterexample makes the limitation of LRH easy to grasp.

- I think the strongest part of the paper is the conceptual separation between what should be predictable and what should not. The distinction between the axis-aligned component, the normal-plane magnitude, and the sector-level phase behavior gives the paper a sharper backbone than a lot of prior steering work. Theorems 4.1 and 4.3 are interesting less because they solve steering and more because they draw a boundary around what one should expect to recover from observable geometry.

- The empirical scope is also fairly broad. Testing 100 concepts across two model families, multiple layers, four steering methods, and multiple token-position strategies is a nontrivial amount of work. The “three implications” structure in Section 6 is a good design choice because it turns a fairly abstract geometric story into specific empirical checks rather than vague post hoc interpretation.

- Among the experiments, Figure 6 is the most convincing to me. The penalty study gives a concrete and intuitive trade-off: larger orthogonal components seem to make concept activation happen sooner, but they also make corruption happen sooner. That is exactly the kind of pattern CRH would predict, and it is stronger evidence than the purely visual probing plots. I also liked that the authors validated the LLM-judge labels against human annotation; given how much of the evaluation pipeline depends on automated labeling, that extra step matters.

Weaknesses

- My main concern is Section 5. The probing results are visually appealing, but the setup risks being circular. The paper optimizes steering vectors, then runs PCA on those optimized vectors, and then interprets the leading components as the cylindrical coordinate system. That makes it hard to tell whether the recovered structure is really a property of the underlying representation geometry or partly a byproduct of the optimization-plus-PCA procedure. A simple control would help a lot here: for example, apply the same pipeline to random vectors with matched norm budgets, or to optimized vectors for a task where no meaningful concept structure is expected, and check whether similar “sector” patterns appear anyway.

- A second issue is the gap between the theory and the experimental substrate. The theory is stated in an idealized output representation space where moving in the space has a direct causal interpretation in terms of outputs, but the experiments are run on intermediate residual-stream states. The paper acknowledges this, but I do not think it resolves it. Right now the reader is asked to accept that the hidden-state geometry is close enough to the idealized output-space geometry for the theorems to remain informative. That may be true, but the bridge is still too informal for how central it is to the paper.

- Related to this, there is a bit of slippage between the 2D “normal plane” introduced in Eq. 7 and the broader orthogonal subspace used later in the appendix proofs and the penalty experiment. In Appendix G.1, the projection operator is effectively defined using the full orthogonal complement of the axis, not the 2D plane from the main text. That does not necessarily kill the intuition, but it weakens the formal neatness of the argument and makes the predictability claim feel less direct than advertised.

- I was also not fully persuaded by the steerability metric in Eq. 17. The paper defines steerability using the slope of a linear fit to
𝑝
(
𝜆
)
p(λ), but the appendix figures show fairly clear nonlinearity, often with a rise-then-fall pattern rather than anything close to linear. If the empirical curve is unimodal, reducing it to a single linear slope seems like a fragile summary statistic. Since Figure 7(a) is one of the paper’s main pieces of evidence for Implication 2, I would want to know whether the result survives alternative choices such as area under the curve, threshold-crossing strength, or peak activation.

- The scale of the experimental evaluation is another limitation. Gemma-2B and LLaMA2-7B are reasonable starting points, but the paper’s story is partly about dimensionality, overlap, and interference. Those quantities may behave differently in larger models with richer representation spaces. Even one 13B+ model would have helped clarify whether the cylindrical picture is robust or mostly a small-to-mid-scale phenomenon.

- Finally, the writing is generally solid, but Section 3.2 is denser than it needs to be. The move from the balancing condition to the definition of the normal plane happens quickly, and the choice of PC1 over the non-target concepts feels more convenient than theoretically compelled. There is also an editing issue in Section 5.1: the “Probing Process” paragraph appears twice in slightly different forms.

---

> ### Author Rebuttal · Authors · 2026-03-29
>
> Thank you for the positive and thoughtful review.
>
> **w1: possible circularity in Section 5 & c1: control outside the PCA-on-optimized-vectors setup.** Thank you for highlighting this central concern. We agree that Section 5 needed a stronger control. We therefore ran a matched random-direction control on Gemma-2B-IT, layer 9, with the same probing budget as the original PCA-based probe. The result is shows that the PCA-based probe shows strong axis and phase structure, while the random-direction control produces a nearly flat loss landscape.
>
> | Metric | PCA-based probe | Random-direction control |
> |---|---:|---:|
> | Global loss mean | 38.72 | 39.80 |
> | Global loss std | 3.43 | 1.16 |
> | Axis loss range | 6.05 | 1.59 |
> | Angle-wise loss variation | 0.51 | 0.07 |
>
> **w2 & c3: bridge from idealized output space to residual-stream states.** Thank you for raising this point. We agree this connection should be clearer.
>
> In practice, steering is often applied to the residual stream, therefore it serves as the validation platform of CRH. In the paper, we believe that,
> if a steering intervention succeeds, the added vector changes the output. This shows that the residual stream already carries the causal signal we want to study. In this case, the output space can be seen as an idealized view of the residual stream.
> We also acknowledge that the residual stream is too noisy to be identical to the output representation space. However, across 100 concepts, we observe consistent CRH patterns from the residual stream. This suggests the residual stream is a reasonable proxy at a statistical level.
> We will revise the paper to make this point clearer and describe the experiments as controlled approximations.
>
> **w3: 2D normal plane versus the full orthogonal complement.** Thank you for pointing this out. We agree this needs to be clearer.
>
> These two are not inconsistent. The difference is that the 2D plane component is not directly observable from \(v\) and \(v_d\), while the full orthogonal component \(v - \mathrm{Proj}_{v_d}(v)\) is observable.
> Our goal in the appendix is to use this observable orthogonal component as a proxy. The argument shows that its variation is aligned with the true plane component, which can therefore be approximated.
> So this is not a change of object, but a proxy for an unobservable quantity. The penalty experiment follows the same idea: it perturbs the full orthogonal component, which also affects the plane component.
> We will revise the paper to make this proxy relationship explicit and avoid confusion.
>
> **w4: steerability metric in Equation 17 & c2: sensitivity of Figure 7(a) to metric choice.** Thank you for raising this point. We think there is a small misunderstanding here.
> Equation 17 is not used for Figure 7(a). Figure 7(a) evaluates Implication 2, which studies the correlation structure with respect to angle and does not depend on the steerability metric in Equation 17.
> Equation 17 is only used for Implication 3, where we compare steerability across vectors.
>
> **w5: scale and larger models & c4: whether the same pattern appears beyond 7B.** Thank you for raising this point.
> Our model choices follow prior work in this area. For example, AxBench evaluates on Gemma-2B and 9B, and the original LRH work mainly studies LLaMA2. As a comparable theoretical study, we believe our experimental scale is in line with existing standards and sufficient to support the main claims.
>
> That said, we agree that larger models are worth checking. Following your suggestion, we ran all three implication tests on **Qwen2.5-14B**.
> We observe similar patterns. For Implication 2, the curve shows the same unimodal trend. For Implication 3, the correlation remains insignificant (Pearson = 0.019, p = 0.297), consistent with our original findings.
> We have added these results to our anonymous GitHub repository (folder: `appl_results_qwen2.5_14b`) for reference.
>
> We will include this result and clarify the scope of our empirical claims in the revision.
>
> **w6:writing clarity, PC1 rationale, and the duplicated probing paragraph.** Thank you for these presentation comments. We agree and will rewrite Section 3.2 to slow down the transition from the balance condition to the 2D normal plane. We will also explain the role of PC1 more directly, as noted above. And thank you for catching the duplicated “Probing Process” paragraph in Section 5.1. We will remove the duplicate and do another careful pass for wording and editing issues.
>
> ---
> *If our clarifications address your remaining concerns, we would greatly appreciate it if you could consider increasing your confidence in the review.*

---

> > ### Author Rebuttal · Reviewer_gBDL · 2026-04-02
> >
> > Thanks for the response. will increase my confidence.

---

> > > ### Author Response · Authors · 2026-04-08
> > >
> > > Dear Reviewer gBDL,
> > >
> > > We're happy to see the concerns have been solved. Again, we sincerely appreciate your continued support and the care you have taken in evaluating our work.
> > >
> > > Best,
> > >
> > > the Authors

---

### Official Review · Reviewer_789j · 2026-03-17

**Soundness:** 3
**Presentation:** 3
**Significance:** 4
**Originality:** 4
**Overall Recommendation:** 5
**Confidence:** 4

**Summary:**

This paper proposes the cylindrical representation hypothesis (CRH), an alternative to the linear representation hypothesis (LRH) that tries to explain why activation steering in LLMs is unreliable at the sample level. The key idea is to decompose representation differences into three components: a central axis (concept direction), a sample-specific normal plane capturing interference from co-occurring concepts, and sensitive vs. non-sensitive sectors within that plane. The authors prove that the normal-plane magnitude is observable (Theorem 4.1) while the sensitive sector's phase is not (Theorem 4.3), and validate these predictions across 100 concepts on Gemma-2B-IT and LLaMA2-7B-Chat.

**Compliance With Llm Reviewing Policy:**

Affirmed.

**Key Questions For Authors:**

1. What does the explained variance look like for the PCA decomposition used to build the cylindrical coordinate system? Specifically, how much variance do the first three PCs capture, and how stable is this across concepts?

2. Have you run any null-model analysis — e.g., applying the same probing pipeline to random vectors or a randomly initialized model? This would be the single most important addition to distinguish genuine geometric structure from methodology artifacts.

3. The theory assumes binary concepts (male/female, present/absent). How would CRH extend to graded or continuous concepts like formality or toxicity? Would the cylindrical geometry still hold?

4. The probing experiments (Section 5) use one-shot optimized steering vectors, but verification (Section 6) uses DiffMean vectors. Could different construction methods yield different cylindrical structures for the same concept? If so, what does CRH describe — the model's intrinsic geometry or the method's geometry?

5. In Equation 14, the exponents m and n are treated as concept-level constants. How variable are these in practice, and do they correlate with anything interpretable about the concepts?

**Limitations:**

The authors discuss some limitations but miss the most important one: the lack of null-model comparisons. They acknowledge restricting to two small models (Gemma-2B, LLaMA2-7B) but do not discuss whether the cylindrical structure would persist at larger scales where representations are higher-dimensional and concepts might be more naturally separable. The reliance on an LLM judge (with only 102 human-validated items) for the steering evaluation is another limitation that deserves more discussion. Societal impact is adequately addressed.

**Strengths And Weaknesses:**

I like the motivation. The sign-flip counterexample in Section 2.2 — showing LRH can't accommodate more than d orthogonal concepts in d dimensions — is a clean argument that immediately justifies why a purely linear view is insufficient. The theoretical results are appropriate for the claims: the observability/non-observability asymmetry between Theorems 4.1 and 4.3 is the paper's key insight. The contradiction argument in Appendix G.3 is clean, and the mixed power-law steerability formula (Eq. 14) leads to the testable prediction validated in Figure 7a.

The restriction to a 2D normal plane (Eq. 7) is interesting. The orthogonal components live in a potentially high-dimensional subspace, and cramming them into 2D necessarily throws away information. How much explained variance do the top-2 orthogonal PCs capture? This number is never reported, which is a notable omission.

---

> ### Author Rebuttal · Authors · 2026-03-29
>
> Thank you for the very positive review. We especially appreciate your recognition of the motivation and the core split between what CRH predicts and what it does not predict.
>
> **c1: explained variance of the PCA cylinder.** Thank you for raising this key diagnostic. We computed the explained variance of the PCA decomposition used in the probing step. The first three PCs explain most of the variance, and this pattern is stable across samples and concepts in the tested setup. We will add these statistics to the appendix and state this point more clearly in the revision.
>
> | Setting | PC1 mean | PC2 mean | PC3 mean | Top-3 mean | Top-3 min | Top-3 max |
> |---|---:|---:|---:|---:|---:|---:|
> | Gemma-2B-IT, layer 9, 2,451 cylinders, 99 concepts | 0.7921 | 0.1191 | 0.0403 | 0.9514 | 0.8520 | 0.9828 |
>
> This shows that the cylindrical coordinate system is not built from a weak low-dimensional summary in this setup. The first three PCs already capture about 95% of the optimized-vector variance on average.
>
> **w1: lack of null-model control & c2: null-model analysis.** Thank you for identifying this as an important missing control. We agree that this analysis is necessary to better separate genuine geometric structure from possible pipeline artifacts. We therefore ran a matched null control on Gemma-2B-IT, layer 9, using the same probing objective and norm budget, but replacing the PCA-derived cylinder directions with random directions. The contrast is clear: PCA-based directions produce much stronger axis and phase structure, while random directions yield a substantially flatter landscape.
>
> | Metric | PCA-based probe | Random-direction control |
> |---|---:|---:|
> | Global loss mean | 38.72 | 39.80 |
> | Global loss std | 3.43 | 1.16 |
> | Axis loss range | 6.05 | 1.59 |
> | Angle-wise loss variation | 0.51 | 0.07 |
>
> These results make the probing evidence more convincing and reduce the concern that the observed sector pattern is only a byproduct of the optimization-plus-PCA procedure. Under matched random directions, the structure becomes much weaker. We will add this control and the corresponding discussion to the revised paper.
>
> **c3: extension to graded or continuous concepts.** Thank you for raising this important question. The current theory is written for binary concepts because this setting gives a clean definition of the sample-specific difference vector and the central axis. At the same time, our probing results suggest a natural local extension to graded concepts: concept activation often changes gradually rather than in one step. In practice, outputs can pass through intermediate states before reaching a clear target expression. We do not want to overclaim this as a formal result, but we agree it is a natural next direction.
>
> | Steering step | Output behavior for concept “Answer in Chinese” |
> |---:|---|
> | 1 | The aurora borealis (north lights) and aurora australis (south lights) are caused...|
> | 9 | 原因 causes jugó北極 lights 的原因：**太陽活動：**北極 scientifiques... |
> | 11 | 光风是导致北极光的主要原因。光...|
>
> **c4:optimized vectors in probing vs DiffMean in verification.** In our view, CRH is the model’s intrinsic sample-specific geometry, not the geometry of one specific steering method. Different vector-construction methods are different estimators of concept-related directions, so they can reveal the same underlying geometry with different fidelity and different noise. This is exactly why the paper separates Section 5 and Section 6. Section 5 uses one-shot optimized vectors because they provide a cleaner local window for visualizing the geometry around a fixed sample. Section 6 then tests observable implications with standard steering vectors used in practice. We will make this division more explicit. We will also clarify that method-dependent differences do not contradict CRH. Instead, they show that some estimators recover the underlying structure more cleanly than others. This interpretation is also consistent with our broader experiments across steering under DiffMean, PCA, etc.
>
> **c5:variability and meaning of m and n in Equation 14.** Thank you for raising this. In the current paper, we do not view the fitted m and n as stable, interpretable semantic descriptors of concepts. Empirically, the best split varies across steering-vector construction methods and settings, and the maximum correlations remain modest. Therefore, Equation 14 mainly serves as supporting evidence for CRH, rather than a practical predictor of steerability. The useful point is the existence of a single best mixed-power split, that is, the unimodal peak, rather than the semantic meaning of the exact exponent values themselves. We will revise the text to make this more precise and to avoid overinterpreting m and n. We also agree that turning this into a practically useful steerability predictor will require more sample-specific information than the current equation uses.
>
> ---
> *Again, thank you for the strong support and careful reading.*

---

> > ### Author Rebuttal · Reviewer_789j · 2026-04-04
> >
> > I was always positive about this paper. Happy to recommend acceptance.

---

> > > ### Author Response · Authors · 2026-04-08
> > >
> > > Dear Reviewer 789j,
> > >
> > > Thank you again for your continued support and for the time and effort you have devoted to reviewing our paper!
> > >
> > > Best,
> > >
> > > the Authors

---

### Decision · Program_Chairs · 2026-04-30

**Decision:**

Accept (regular)

**Comment:**

This paper introduces the Cylindrical Representation Hypothesis (CRH) as an alternative to the Linear Representation Hypothesis (LRH) for understanding activation steering in large language models. The key idea is to decompose representation differences into a central axis (capturing the concept direction), a sample-specific normal plane (capturing interference from co-occurring concepts), and sensitive versus non-sensitive sectors within that plane. The paper establishes analysis for the distinction between what aspects of this geometry are observable and what are fundamentally unobservable, providing an explanation for why steering is reliable in aggregate but brittle at the sample level. These predictions are validated through experiments.

All reviewers find the paper interesting, recognize its contributions in understanding internal representations, and recommend acceptance. I concur with their assessment and encourage the authors to revise the paper to incorporate the additional experiments and discussions raised during the rebuttal.